# Quality transformation of dissolved organic carbon during water transit through lakes: contrasting controls by photochemical and biological processes

Martin Berggren[1], Marcus Klaus[2], Balathandayuthabani Panneer Selvam[1], Lena Ström[1], Hjalmar Laudon[3], Mats Jansson[2], Jan Karlsson[2]

[1]Department of Physical Geography and Ecosystem Science, Lund University, SE-223 62, Lund, Sweden
[2]Department of Ecology and Environmental Science, Umeå University, SE-90187, Umeå, Sweden
[3]Department of Forest Ecology and Management, Swedish University of Agricultural Sciences, SE-90183, Umeå, Sweden

*Correspondence to*: Martin Berggren (martin.berggren@nateko.lu.se)

**Abstract.** Dissolved organic carbon (DOC) may be removed, transformed or added during water transit through lakes, resulting in qualitative changes in DOC composition and pigmentation (color). However, the process-based understanding of these changes is incomplete, especially for headwater lakes. We hypothesized that because heterotrophic bacteria preferentially consume non-colored DOC, while photochemical processing remove colored fractions, the overall changes in DOC color upon water passage through a lake depend on the relative importance of these two processes, accordingly. To test this hypothesis we combined laboratory experiments with field studies in nine boreal lakes, assessing both the relative importance of different DOC decay processes (biological or photo-chemical) and the loss of color during water transit time (WTT) through the lakes. We found that influence from photo-decay dominated changes in DOC quality in the epilimnia of relatively clear headwater lakes, resulting in systematic and selective net losses of colored DOC. However, in highly pigmented brown-water lakes (absorbance at 420 nm >7 m$^{-1}$) biological processes dominated, and there was no systematic relationship between color loss and WTT. Moreover, *in situ* data and dark experiments supported our hypothesis of selective microbial removal of non-pigmented DOC, mainly of low molecular weight, leading to persistent water color in these highly colored lakes. Our study exemplifies that brown headwater lakes may not conform to the commonly reported pattern of selective removal of colored constituents in freshwaters, as the DOC can show a sustained degree of pigmentation upon transit through these lakes.

## 1 Introduction

The color of water is a defining feature of freshwater ecosystems, primarily caused by inputs of brown-pigmented dissolved organic carbon (DOC) from terrestrial runoff (Xiao et al., 2015). Recent concerns have been raised of widespread increases in color and DOC concentrations in the northern hemisphere, caused by a combination of factors involving a warmer climate (Lepistö et al., 2014;Pagano et al., 2014), an intensified hydrological cycle (Weyhenmeyer et al., 2012;Fasching et al., 2016) and release of DOC that previously was immobilized in soils due to acidification (Monteith et al., 2007). This rise in colored DOC, reviewed by Solomon et al. (2015), is predicted to reduce aquatic productivity (Karlsson et al., 2009), change food webs

and population structures (Jansson et al., 2007), alter the stoichiometry and magnitude of bioavailable nutrients pools (Berggren et al., 2015b), and cause increased freshwater $CO_2$ outgassing (Lapierre et al., 2013). Thus, water color is key to understanding fundamental aspects of aquatic ecosystem functioning in a changing environment.

Inland waters represent a significant component in the global carbon cycle, e.g. emitting greenhouse gases to the atmosphere at the rate of at least 1 or 2 Pg C per year (Raymond et al., 2013;Cole et al., 2007). A fundamentally important water-column process that generates carbon dioxide ($CO_2$) is the microbial degradation of terrestrially-derived DOC (Lapierre et al., 2013;Fasching et al., 2014). Significant amounts of DOC can also be mineralized by ultraviolet (UV) sunlight in lakes (Koehler et al., 2014) and running waters (Cory et al., 2014). However, while much research attention has been drawn to the $CO_2$ production from these different processes, surprisingly little is known about the relative role played by biological and photochemical processes for DOC quality transformations and, in particular, for the removal of color.

On large scales, color tends to decrease faster than DOC along the land-sea continuum (Weyhenmeyer et al., 2012), partly because non-colored DOC might be added by algae in productive waters (Creed et al., 2015), but the circumstances allowing for preferential net losses of colored DOC in unproductive lakes are unclear. Some studies have reported relative losses of colored DOC across lake basins with increasing theoretical residence times (Köhler et al., 2013;Curtis and Schindler, 1997), while other studies have found preferential loss of non-colored DOC in laboratory biodegradation experiments (Hansen et al., 2016) and in time-series analyses of brown headwater lakes (Berggren et al., 2009). Although bacteria do consume colored humic substances at low rates (Tranvik, 1988), the biological degradation of DOC is unlikely a mechanism leading to selective color loss because bacteria tend to consume non-colored DOC (Asmala et al., 2014;Hansen et al., 2016). An exception is the apparent preferential use of organo-ferric colloids by bacteria, where the removed color comes from iron, not DOC (Oleinikova et al., 2017). The UV light oxidation could theoretically explain losses of colored DOC and production of non-colored DOC (Stubbins et al., 2010;Kellerman et al., 2014), but efficient photo-processing has been found mainly in relatively DOC poor water (Molot and Dillon, 1997) and in alkaline lakes (Reche et al., 1999), and not systematically in unproductive DOC-rich lakes (Amon and Benner, 1996;Molot and Dillon, 1997;Jonsson et al., 2001). Thus, the processing of colored DOC remains poorly understood in response to water transit through typical unproductive DOC-rich headwater systems (Weyhenmeyer et al., 2014).

Most of the studies that have addressed changes in water chemistry in response to water transit times in lakes have applied fixed theoretical mean residence time values (Köhler et al., 2013;Curtis and Schindler, 1997;Weyhenmeyer et al., 2012). However, in reality water transit times through lakes and reservoirs vary several-fold over time, as a result of temporal flow variations (Li et al., 2015;Rueda et al., 2006). Thus, the processing of DOC in response to the actual water transit time (WTT; average time spent by the water molecules in the lake) through a given lake has been overlooked, addressed only in few studies (Berggren et al., 2009;Berggren et al., 2010b). For this approach to be successful, the DOC needs to enter a lake in distinct pulses, each time with similar concentrations and chemical properties, such that subsequent DOC quality changes in the lake are dependent on the WTT. Nonetheless, WTT assessment stands out a tool that can help fill in the knowledge gap on the hydrological and biogeochemical mechanisms behind DOC quality and color change in headwater lakes.

In this study we therefore combine WTT calculations for a range of different headwater lakes with laboratory simulations of DOC processing to determine the relative importance of different processes that remove DOC (biological or photo-chemical) and color over time. We chose lakes located within a 50-70 km radius around the Krycklan catchment in northern Sweden, where there is extensive research in support of the assumption that DOC export is highly episodic, with

pulses that generally bring DOC of similar quantity and quality during peak flow (Laudon et al., 2011). We hypothesized that heterotrophic bacteria preferentially consume non-colored DOC fractions, resulting in small overall color loss during water retention in lakes where bio-degradation represents the dominant DOC transformation process. On the contrary, selective loss of colored DOC could be expected in lakes where the DOC transformation is dominated by photo-chemical processing.

## 2 Methods

### 2.1 Study design overview

Targeting the well-studied Björntjärnarna brown-water catchment in northern Sweden (Berggren et al., 2009;Berggren et al., 2015a;Karlsson et al., 2012), a large data set for a single catchment was first compiled and analyzed. In total 260 samples were obtained over seven study years from the inlets, epilimnia, hypolimnia and terminal outlet of two tightly connected lakes called 'Övre Björntjärnen' and 'Nedre Björntjärnen'. These chain lakes share 92% of the catchment area, and their epilimnia are

tightly connected by a short (70 m length) stream, making the same water pass through the two lakes in sequence. The purpose of this first analysis was to analyze one large pooled data set to maximize power, i.e. the chance of finding significant patterns in the response variables with increasing WTT.

For a second part of the analysis, we collected a limited number of samples (ca 10 per year and site) during 3-4 years from seven additional 'survey lakes' along a gradient of DOC and color, increasing the total number of study lakes to nine.

This data were used to increase the representativeness of the study, given the variations in UV light exposure in the water column and possible differences in UV light degradation between brown-water and clear-water lakes. We selected unproductive boreal lakes (Chl-a < 2 $\mu$g L$^{-1}$) because the majority of lakes are located in the boreal region (Verpoorter et al., 2014), where nutrient concentrations are low yet lake DOC concentrations and optical conditions vary widely (Karlsson et al., 2009). In the analyses of these survey lakes, epilimnetic and hypolimnetic data were kept separate, due to the differences in

light climate with different water depths. To understand the relative role played by biological and photo-chemical processes in the qualitative transformations of DOC, we performed laboratory bioassay experiments on water from three of the lakes (Table 1), where changes in optical water properties were measured in dark bacterial bioassays and under UV light irradiation, respectively. Since the two processes had systematically different impact on DOC optical properties, we could use optical indices to see which of the processes that dominated based on how the indices changed *in situ* with increasing natural WTT in

the lakes.

We assumed no significant variations in color induced by iron (Fe) or pH in this study. Total unfiltered Fe concentrations are in the order of one mg L$^{-1}$ at the inlets, epilimnia and outlets (personal communication, D Bastviken,

Linköping University; measured in the four most colored lakes). Although the abundance and speciation of Fe can affect optical properties of freshwaters (Pullin et al., 2007), the effect on absorbance in the water should be marginal (Weishaar et al., 2003). According to Kritzberg and Ekström (2012), the contribution from one mg $L^{-1}$ of Fe to absorption coefficient at 420 nm ($a_{420}$) is 0.8 $m^{-1}$, corresponding to 5-10% of the observed $a_{420}$ in the four lakes that have been analyzed for total Fe. Regarding pH, Pace et al. (2012) showed that the absorption coefficients increases sharply beyond a pH of 7 due to changes in the three-dimensional structure of the DOC molecules. The absorption at low UV wavelengths in particular can also be enhanced by an extremely low pH (~4), potentially leading to higher photo-reactivity (Anesio and Granéli, 2003;Gennings et al., 2001), although these effects appear relatively small compared to those at high pH (Pace et al., 2012). In this study, the observed range in pH across all sites and sampling dates was 3.4-6.8 (mean = 5.4, $SD$ = 0.6), which is below the threshold for major pH interference according to Pace et al. (2012).

## 2.2 Study site descriptions

The nine Swedish boreal lakes that were selected (5-10 m max depth) varied more than 5-fold in color and ca 3-fold in DOC (Table 1). All lakes have previously been depth-profiled using an echo sounder. Regional mean annual temperature is 1˚C and the average annual precipitation is 500-600 mm, of which half arrives as snow. Lake ice is generally present from late October to early May. While the lakes have small areas, varying only between 0.01-0.05 $km^2$, the catchment areas vary 100-fold from 0.03 $km^2$ to 3.25 $km^2$, resulting in mean epilimnetic water transit times from 0.2-0.3 yrs to ca 1-3 yrs in the lakes with the largest and the smallest watersheds, respectively (Table 1). The catchments are mainly (>75%) covered by coniferous forest (*Picea abies, Pinus sylvestris*) and *Sphagnum*-dominated mires (<25%). Forests are managed and have widely varying age, both within and between the catchments, from 5-100 years. Location, mean optical properties and DOC concentrations of each of the study lakes are presented in Table 1. The four clearest lakes have no permanent inlet streams, their catchments are very small (Table 1) and they are surrounded by forested hills. The five browner lakes receive hydrological input from streams and they are located in relatively flat lower parts of the catchments, with abundant peatlands in their direct surroundings.

Three of the lakes (Table 1) received inorganic N additions 2012-2014 (as part of another study) to create a slightly elevated nitrate concentration, by 0.1 mg N $L^{-1}$ (Klaus et al., 2017). This fertilization could potentially affect the DOC decay(Berggren et al., 2007). Therefore, before initiation of the lake fertilization, we tested the potential influence of N by performing 2-week *in vitro* DOC and color loss measurements (bioassays) on water from the lake Nedre Björntjärnen, which has the lowest natural ratio of inorganic N to DOC among the nutrient amended lakes, and thus would be at the highest risk for bias. The bioassays were performed as described previously in detail (Berggren et al., 2009) at 20°C dark conditions with ambient bacterial communities and with additions of single spikes (1 mg N $L^{-1}$ added) of ammonium nitrate at the beginning of the incubations, using epilimnetic water obtained in winter, spring, summer and fall. We found no N additions effect on DOC degradation or change in water color during these bioassays experiment (Fig. S1). Moreover, none of the N-amended lakes appeared as outliers in this study. Therefore, we assume that the lake N addition had no critical impact on the results or the conclusions drawn in this work.

**2.3 Sampling and water analysis**

Sampling was carried out between 2006 and 2014 (3-7 years of data per lake). Water from epilimnion and hypolimnion (mid-depths or composite samples) was collected every 2-3 weeks throughout the ice-free seasons, and occasionally under ice, at the deepest point of each lake (see sample numbers in Table 1). Additional samples for the detailed analysis of the Björntjärnarna catchments were obtained on most sampling dates at the headwater inlet and outlets of the chain lakes (Björntjärnarna). Sampled water was stored in cooling boxes until processing in the lab within 2-10 hours. Temperature profiles were obtained with electronic sensors at each sampling occasion (plus occasional additional dates) and used to calculate volumes above and below the thermocline depth, defined as the mid-depth of the transect where temperature changed >1°C m$^{-1}$. Discharge was assessed as described in Supplementary information, Text S1.

In the laboratory, lake water was filtered with acid-washed 0.7 µm glass fiber filters (Whatman GF/F). Absorbance spectra of the filtrate were measured at room temperature in 1 cm quartz cuvettes using a Jasco V-560 UV-VIS spectrophotometer. Blank values from deionized water were subtracted from the spectra. An aliquot of 40 ml of the filtrate was acidified (50 µL 1.2M HCl) and stored in darkness at 6˚C until DOC analysis by high-temperature catalytic oxidation using a HACH-IL 550 TOC-TN analyzer (Hach-Lange GmbH Düsseldorf, Germany). In the Björntjärnarna chain lakes, water from 12 dates in 2009 was analysed for organic acids, free amino acids and simple carbohydrates using a liquid chromatography-ion spray tandem mass spectrometry (LC-MS) system. The LC-MS system consisted of a Dionex (Sunnyvale, CA, USA) ICS-2500 liquid chromatography system and an Applied Biosystems (Foster City, CA, USA) 2000 Q-trap triple quadrupole mass spectrometer. The method is described in further detail in Ström et al. (2012).

**2.4 Water transit time assessments**

The transit time of the water volume that resides in a lake at a given moment is dependent upon the retention and renewal history of that water. We assumed that WTT increases with +1 per unit of time and that it decreases in proportion to how the water volume (Vol$_{total}$) is replaced by new inflowing water (Flow rate; volume per unit time), which gives a change in WTT per unit time (dWTT/dt) according to Eq. 1.

$$\frac{dWTT}{dt} = 1 - WTT * \frac{Flow\ rate*dt}{Vol_{total}} \tag{1}$$

However, this continuous function (Eq. 1) is not suited to be applied directly in this study, because our data is discrete and further involves two depth strata (epilimnion and hypolimnion) with reciprocal entrainment effects due to dislocation of the thermocline. Therefore we adapted discrete functions for the changes in epilimnetic and hypolimnetic WTT from one day (t) to the next day (t+1) (Berggren et al., 2010b;Berggren et al., 2009). Using these functions (Eq. 2-5), lake WTTs were calculated iteratively for each day in sequence from discharge (measured daily) and lake volume data (epilimnetic and hypolimnetic; daily values obtained by linear interpolations between sampling dates). To get realistic WTT values for the first day of the

study period, the iteration was initiated from a date 10 years in advance. An arbitrary WTT starting value could then be chosen without impact on the calculated WTTs of the study years. For the pre-study period, the mean seasonal mixing pattern for each lake (see Fig. S2a) was used to generate daily epilimnetic and hypolimnetic proxy volumes.

For any given day (t), a certain volume of inflowing water ($Vol_{inflow}$) was considered to mix with a certain volume of
epilimnetic lake water ($Vol_{epi}$; above mid-thermocline depth) and, during days with downward dislocation of the thermocline, with an additional volume of hypolimnetic water ($Vol_{hypo\ flow}$). Thus, after the day in question (t+1), the resulting new mean WTT equals the volume weighted average WTT of these different volumes that mixed during day t. In addition, the transit time also changes with '+1' per unit of time, as the water resides 1 d in the lake during the day (t). Hence the resulting WTT on day 't+1' was calculated according to Eq. 2 (days without downwards thermocline dislocation) or Eq. 3 (days with
downwards thermocline dislocation). $Vol_{hypo\ flow}$ (t) was given from the decrease in the hypolimnetic volume from day t to the next day (t+1).

$$WTT_{epi}(t + 1) = 1 + \frac{WTT_{in}(t) \times Vol_{inflow}(t) + WTT_{epi}(t) \times Vol_{epi}(t)}{Vol_{inflow}(t) + Vol_{epi}(t)} \qquad (2)$$

$$WTT_{epi}(t + 1) = 1 + \frac{WTT_{in}(t) \times Vol_{inflow}(t) + WTT_{epi}(t) \times Vol_{epi}(t) + WTT_{hypo}(t) \times Vol_{hypo\ flow}(t)}{Vol_{inflow}(t) + Vol_{epi}(t) + Vol_{hypo\ flow}(t)} \qquad (3)$$

After a day with reduced thermocline depth, the new hypolimnetic WTT (t+1) resulting from entrainment of epilimnetic water ($Vol_{epi\ flow}$) into the hypolimnion ($Vol_{hypo}$) was calculated using Eq. 4. Again, the transit time also changes with '+1', as the water resides in the lake during the day in question. After a day without reduced thermocline depth, the $WTT_{hypo}$ was unaffected
by mixing with epilimnetic water (Eq. 5). $Vol_{epi\ flow}$ (t) was given by the increase in the hypolimnetic volume from day (t) to the next day (t+1).

$$WTT_{hypo}(t + 1) = 1 + \frac{WTT_{epi}(t) \times Vol_{epi\ flow}(t) + WTT_{hypo}(t) \times Vol_{hypo}(t)}{Vol_{epi\ flow}(t) + Vol_{hypo}(t)} \qquad (4)$$

$$WTT_{hypo}(t + 1) = 1 + WTT_{hypo}(t) \qquad (5)$$

Inflowing water from the catchment ($Vol_{inflow}$) was assigned the WTT of 0, as the headwater inlet streams represent water transit times that can be considered negligible compared to the WTT in the lakes (Berggren et al., 2009). Moreover, there were no upstream lakes in the catchments, in addition to the study lakes themselves, implying that the drainage represented true
headwater sources directly from surrounding soils. However, in the lower chain lake (Nedre Björntjärnen), the WTT of inflowing water was considered to equal the WTT of outflowing (epilimnetic) water from the upper chain lake (Övre

Björntjärnen). Similarly, we assumed that WTT of outflowing water from the lower chain lake equal the WTT for epilimnetic water in this lake.

Our consideration that all inflowing water mixed with the epilimnion (not hypolimnion) is supported by the fact that all of the lakes with permanent inlets (five of the study lakes) have inlet streams entering shallow areas of the respective lakes, i.e. without hypolimnia, where the water is forced to mix with the epilimnion. Thus even if the inflowing stream water sometimes had a temperature (and thus density) similar to that of the hypolimnion, no down welling was likely to happen.

## 2.5 Response variables

We used the decadic absorbance coefficient $a_{420}$ ($m^{-1}$) as a measure of 'color', conventional in water monitoring and research in the study region (Kritzberg and Ekström, 2012;Weyhenmeyer et al., 2012). Besides DOC and the color indicator $a_{420}$ (Weyhenmeyer et al., 2012), we analyzed two optical ratios that indicate qualitative changes in the dissolved organic matter in response to changing WTT. Firstly, we used the absorbance ratio $a_{254} : a_{365}$, which describes a shift towards absorption in the red part of the spectrum, and thus tends to be negatively related to average molecular DOC weight (Dahlén et al., 1996) and positively correlated to low molecular weight DOC compounds (Berggren et al., 2010a). There are many other spectral slope indices in the literature essentially providing the same information, but we chose $a_{254} : a_{365}$ since it is a simple index that has been used previously in the study area (Berggren et al., 2007;Ågren et al., 2008a). The direction of change in this ratio is indicative of the dominant DOC transformation process: $a_{254} : a_{365}$ increases with UV light processing (Dahlén et al., 1996), but it decreases in response to bacterial DOC processing (Berggren et al., 2007). In agreement with the expectations based on Berggren et al. (2010a), $a_{254} : a_{365}$ in this study was positively correlated to both absolute (mg C $L^{-1}$) and relative (% of DOC) total concentrations of low molecular weight carbon compounds in the form of organic acids, free amino acids and simple carbohydrates (Fig. S3).

Secondly, we used the ratio $a_{420} : DOC$. If this ratio increases with WTT, then non-colored DOC is selectively removed (or more colored DOC added), but if $a_{420} : DOC$ decreases, then this is either due to selective decay of colored DOC (Weyhenmeyer et al., 2012) or selective addition of low-pigmented DOC (Creed et al., 2015). It should be mentioned here that also other indices of color per unit DOC are common in the literature, especially specific UV light absorbance at 254 nm ($SUVA_{254}$). However, in our study the overall relationship between $SUVA_{254}$ and $a_{420} : DOC$ was strong and linear ($r^2 = 0.80$, $n = 680$; all sites and sampling dates), and the two variables showed the same patterns in response to changing WTT. Therefore, to avoid presenting the same patterns twice, we do not report $SUVA_{254}$ in this paper.

## 2.6 Laboratory experiments

We performed laboratory experiments on water from three catchments to disentangle the isolated effect on DOC quality by UV light degradation from that of microbial processing. The purpose with the experimental design was to create conditions during which either 1) photochemical reactions strongly and dominantly influenced the DOC transformation, or 2) microbial degradation strongly dominated the DOC transformation. Therefore, we measured the DOC quality responses to a large light

dose or a long microbial process time in the dark. However, the experiments were not designed to reflect *in situ* decay rates. For example, temperatures and likely also microbial communities were different in the bottle experiments compared to the field.

In the dark microbial process experiment, we used water either from the lake inlets or from the epilimnia at times when the water had only resided a short period in the lake according to our model (< 1 month on average). The unfiltered natural water samples (with ambient microbial community) were then incubated in darkness for 450 days at 20 °C in 1 L acid-washed Duran glass bottles, with a gas headspace (~100 mL) containing sufficient $O_2$ to theoretically oxidize all DOC in the samples. The selected incubation time (450 d) was similar to the mean WTT in the study lakes (about 1 yr on average; Table 1). Although protozoa were not removed, it can based on Daniel et al. (2005) be assumed that bacteria vastly dominate the biomass (~90%) when dark bioassays are performed on natural humic water. We assumed that there was no nitrogen or phosphorus limitation due to the high concentrations of DOC (~15 mg $L^{-1}$), known to be associated with high organic nutrient bioavailability in lakes (Soares et al., 2017) and streams (Jansson et al., 2012) of the region. Our assumption is supported by Jansson et al. (2001), who showed that bacterial metabolism in humic lakes is generally not nutrient limited when DOC is higher than ~15 mg $L^{-1}$.

For the light experiments, we chose a slightly different approach. While the microbial processing happens continuously in the entire water column of the lakes, UV light processing occurs only in a thin superficial layer of the lakes, during daytime, and ice free conditions. This means that, although photo- and bio-degradation can happen simultaneous, much of the DOC likely has stayed a variable and potentially long time in darkness before getting in contact with UV light (Gonsior et al., 2013). For this reason, we used water that had first been incubated with microbes in the dark (as described above) as starting material for the UV light experiments. Hence, 10 ml filtered (0.45 μm) samples were incubated for 24h in cylindrical quartz vials placed horizontally on a spinning disc (0.67 rpm) in a 20°C climate chamber, at ca 40 cm distance from two xenon-sodium lamps. The UV irradiation of the different parts of the disk was within 3.64-6.89 W $m^{-2}$ for UV-A and 0.06-0.1 W $m^{-2}$ for UV-B according to radiation measurements (Spectroradiometry, International Light Inc.). Spinning of the disk ensured equal light dose received by all samples. Based on Bertilsson and Tranvik (2000), we used complete spectra for lamp irradiation and absorption to calculate that the samples absorbed a UV light dose of approximately 100 MJ $m^{-3}$, equivalent to at least two years of water column-integrated *in situ* UV light absorption. The bacterial DOC processing during irradiation was considered negligible, because the rate of DOC loss under light was 1-2 order of magnitudes higher than in dark incubations with microbial degradation only.

The dark and light experiments were performed on 14 samples from Övre Björntjärnen, Lillsjöliden and Struptjärnen (4-5 per site, from spring to fall). The DOC and absorption coefficients were analyzed before and after each experiment, and the changes from the beginning to the end of the incubation (final value minus start value) were calculated for the absorbance ratio $a_{254} : a_{365}$ and the carbon color indicator for $a_{420} : DOC$. In addition, the relative change (%) in color ($a_{420}$) from beginning to end of the incubations was calculated and used for a qualitative comparison with loss in $a_{420}$ during water transit through the different lakes *in situ*.

**2.7 Statistics**

The response of spectrophotometric variables to changing WTT was evaluated through linear mixed effects regression (LMER) in the pooled dataset from the Björntjärnarna chain lakes, where WTT impact was considered a fixed effect and 'site' was added as random factor to allow for potential differences in the relationships (different intercepts only) between the monitoring stations. However, ordinary linear regression was used in the individual analyses of the survey lakes. Temporally, all response variables showed systematic and significant ($p < 0.05$) autocorrelation for a time lag of 1 step, i.e. the 2-3 week sampling frequency (Box-Ljung autocorrelations around 0.5; software IBM SPSS 22), but there was no systematic autocorrelation for two time steps, i.e. 4-6 weeks. In order to not grant significance too generously, considering the temporal autocorrelation, we adjusted the α for relationships between WTT and response variables from 0.05 to 0.01. According to standard significance tables, it takes roughly twice as many observations to obtain significance of a correlation at the 0.01 level, compared to the 0.05 level, implying that an α adjustment from 0.05 to 0.01 approximately takes into account that only every second observation in the time-series could be assumed to be independent. Although it would be possible to explicitly include an autocorrelation term in the LMER models, the α adjustment was chosen since it could be applied in a systematic way to all regression results in the study, i.e. both to the LMER models and to the ordinary linear regression models. The 0.05 level was considered only marginally significant.

For the laboratory experiment results, we used 2-tail paired t-tests to test for changes between initial and ending conditions. All statistics were performed using IBM SPSS 22, except for the linear mixed effects regressions performed using the statistical package 'lm4' for R. To obtain conditional and marginal $R^2$ estimates ($R^2c$ and $R^2m$, respectively) for mixed models, the package 'MuMIn' was used, while significance of coefficients and intercepts (fixed effects) were tested with the package 'lmerTest'.

**3 Results**

**3.1 Seasonal water transit time patterns**

The different study lakes showed a coherent seasonal mixing pattern, with stable thermal stratification from mid-May to mid-September, in between of spring and fall overturns (Fig. S2). The temporal variability in WTT was partly controlled by annually recurring high-flow events that caused systematic drops in the WTT, by approximately 0.2-0.5 years in spring (due to snow melt) and 0.1-0.2 years in fall (due to rain storms). Conversely, during low flow in winter and summer all sites showed a slow but stable increase in WTT (Fig. S2). For all catchments, 50% or more of the annual runoff was represented by discharge above the 90[th] percentile, i.e. a majority of the discharge happened during hydrological episodes.

In the site-by-site analysis of survey data from all nine lakes, the water transit time spanned from 0.2-3.1 years for epilimnetic samples from each respective site (mean across all sampling dates) and 0.3-3.7 years for the corresponding hypolimnetic samples. Within each lake, there was considerable variability in WTT, with on average twice as high epilimnetic

WTT after the driest periods compared to the wettest periods. The temporal span in WTT, i.e. highest minus lowest value for each site, varied from 0.4 years to 1.8 years.

## 3.2 Björntjärnarna chain lakes

In the pooled data set from the Björntjärnarna brown-water chain lakes, we found significant relationships (linear mixed effects regression) between spectrophotometric response variables and the WTT. The ratio $a_{254}$ : $a_{365}$, which indicates relative abundance of low molecular weight DOC, showed a decreasing trend (from 4.1 to 3.8) over the span from 0 to 0.8 years of WTT in this lake system ($R^2m = 0.15$, $R^2c = 0.19$, $n = 260$; Fig. 1a). At the same time, the DOC became relatively more colored, demonstrated by the significant positive trend for the ratio $a_{420}$ : DOC ($R^2m = 0.16$, $R^2c = 0.20$, $n = 260$; Fig. 1b). For changes in DOC (Fig. 1c) there was no significant fixed effect caused by water transit time, although the random factor (site) explained 17% of the variance (difference between $R^2c = 0.17$ and $R^2m = 0.00$), largely due to higher DOC at the inlet than in the lakes. The overall color (absorbance at 420 nm) remained remarkably constant with increasing WTT (Fig. 1d).

## 3.3 Survey lakes

The ordinary linear regression results on data from epilimnia and hypolimnia of the nine survey lakes are detailed in Table S1, shown with complete data plots in Fig. S5 and summarized in Fig. 2. These results partly conformed to the patterns in Björntjärnarna, as three out of the nine epilimnetic sites showed significant ($p < 0.01$, two cases) or marginally significant ($p < 0.05$, one case) linear decreases in $a_{254}$ : $a_{365}$ with increasing WTT (Fig. 2a; Table S1). In one case this trend was significant at the 0.01 level both above and below the thermocline. On the contrary, in the epilimnia of three clearer lakes, epilimnetic $a_{254}$ : $a_{365}$ increased with increasing WTT (Fig. 2a). Furthermore, only two of the most colored lakes showed increases in $a_{420}$ : DOC with increasing WTT significant at the 0.01 level (Fig. 2b; Table S1), observed both above and below the thermocline in one case. On the contrary, the epilimnion of one of the lakes showed decreasing $a_{420}$ : DOC over the WTT gradient (Fig. 2b). Interestingly, the DOC quality in lakes with intermediate color appeared less responsive to changes in WTT than DOC quality in clearer or browner lakes, exemplified by complete lack of relationships between DOC properties and WTT in the intermediate Lake Lillsjöliden (Table S1, Fig. S5).

When analyzing the nine lakes one by one, significant ($p < 0.01$) losses of DOC with increasing WTT were found only in two of the lakes with intermediate color (Fig. 2c). Unexpectedly, one lake (Stortjärnen) showed a significant trend of increasing DOC with increasing WTT, indicating release of DOC from within the lake or its benthic/littoral contact surfaces. In terms of the overall color of the lakes (absolute $a_{420}$ values), significant ($p < 0.01$) increases with increasing WTT were found in two hypolimnia and one epilimnion (Fig. 2d). In the epilimnetic water of the clearer lakes, the $a_{420}$ tended to decrease, but no significant pattern of decreasing $a_{420}$ was found for hypolimnetic sites (Fig. 2d).

We further found that there were systematic shifts in the dynamics of both $a_{254}$ : $a_{365}$ and the index $a_{420}$ : DOC along the gradient of increasing mean color of the different lakes. In fact, the rate of change in epilimnetic $a_{254}$ : $a_{365}$ per unit WTT was strongly negatively related to the mean $a_{420}$ of the respective lakes ($R^2 = 0.94$, $n = 9$, $p < 0.001$; Fig. 3a). Furthermore, the

rate of change in epilimnetic $a_{420}$ : DOC per unit WTT was positively related to the mean $a_{420}$ ($R^2 = 0.69$, $n = 9$, $p < 0.01$; Fig. 3b).

### 3.4 Experiments

To understand the mechanisms behind such patterns, we performed laboratory light and dark experiments. In the light treatment, the changes in the $a_{254}$ : $a_{365}$ ratio (significantly positive) and in $a_{420}$ : DOC (non-significant slightly negative) were similar to the changes observed as functions of WTT in clear epilimnetic waters (Fig. 3c-d). In the dark bacterial bioassays, the changes in both of these ratios were significant in the opposite direction relative to the light treatment (Fig. 3c-d), i.e. similar to the changes observed in brown epilimnetic waters (Fig. 3a-b).

### 3.5 Overall color loss

Finally, we multiplied the *in situ* rate of epilimnetic color loss in the survey lakes (slopes ± SE of epilimnetic $a_{420}$ regression models; see Table S1) with the mean WTT for the respective sites (Table 1) to find out how much total change there was in water color upon transit through each lake. The change was then expressed as a proportion (%) of the mean WTT for respective lakes. This showed that losses corresponding to 19-79% of mean $a_{420}$ occurred in the four clearest lakes, whereas the $a_{420}$ either showed no change or increased slightly in the five brownest lakes (Fig. 4). These changes in color upon lake transit largely overlapped the ranges of color loss shown in the light (22-51% loss) and dark (13% increase to 36% loss) incubation experiments, respectively. The change in color upon transit through the lakes over the gradient of increasing mean $a_{420}$ was best described (in terms of fit) by a logarithmic curve (Fig. 4).

## 4 Discussion

### 4.1 Main findings

Our results show that brown headwater lakes do not necessarily conform to the generally reported patterns of efficient and selective removal of colored constituents in freshwaters. Based on a rigorous data set from the Björntjärnarna chain lakes ($n = 260$) spanning seven years of measurements, we found that the color ($a_{420}$) was sustained at a constant level over water transit times from zero up to 0.8 years. In these lakes the major discharge pulses alone (snow melt and storms) renew all the water at least once per year, and the processes that removes colored DOC are too slow to result in significant color loss in between of these discharge pulses. Thus, even if input of humic materials from the catchment represent a relatively photo-reactive DOC source (Lindell et al., 2000;Vachon et al., 2016), the photo-bleaching in Björntjärnarna apparently was not sufficient to cause a significant net loss of color. Moreover, the ratio between $a_{420}$ and DOC increased significantly over time, likely due to preferential microbial consumption of non-colored low molecular weight DOC, supporting our hypothesis.

In the multi-lake comparison our results demonstrate contrasting DOC quality dynamics for different types of lake ecosystems. We argue that changes in the DOC quality indices $a_{420}$ : DOC and the absorption ratio $a_{254}$ : $a_{265}$ strongly indicate

that microbial processes dominated the DOC quality transformation in the brownest lakes, while photochemical processes dominated in the clearest lakes. Epilimnetic waters of relatively clear lakes showed losses in $a_{420}$ with WTTs, a pattern also found elsewhere in Sweden (Müller et al., 2013;Weyhenmeyer et al., 2012). Lakes of intermediate color were however relatively irresponsive to changes in WTT, suggesting that the DOC quality and color in these lakes would be less sensitive to
hydrological events such as rainfall or drought.

## 4.2 Possible bias

A factor that could potentially bias interpretations of how DOC properties change in response to WTT is that the chemistry of source water is variable over time, leading to seasonal changes in the concentration and quality of inflowing DOC to the lakes (Ågren et al., 2008b). For example, during low flow it has been shown that headwater streams in the region can have unusually
high concentrations of colored wetland-derived DOC (Laudon et al., 2011). However, in our measurements of DOC concentrations and properties in the inlet stream to the Björntjärnarna catchment (Fig. S4), we found no systematic patterns with discharge. Moreover, headwater sources in general show relatively small variability in water chemistry during episodic flow, which represent a majority of the annual DOC export from small catchments (Laudon et al., 2011) and thus also the majority of the DOC which was processed in the study lakes during subsequent low flow periods. Similar to what has been
reported elsewhere for headwaters (Wilson et al., 2013;Boyer et al., 1997), a majority of the discharge happened during peak flow (>90[th] percentile flow rate) and the low flow periods played a negligible role (Fig. S4). Therefore, it appears unlikely that the patterns in DOC and optical properties with increasing WTT in the lakes would be primarily driven by temporal variations in inflowing water.

     Another potential bias is represented by new sources of DOC, i.e. other than catchment runoff, which can contribute
to the development of lake color over time (Creed et al., 2015). For example, DOC can be added internally within lakes by release from algae, sediments, macrophytes or littoral peats and marshes in direct contact with the lake water (Wetzel, 2001). The lakes in this study are unproductive with minor contributions from primary producers to the DOC pool, exemplified by their negligible role of in a previous organic carbon budget for Övre Björntjärnen (Karlsson et al., 2012). However, one of the study lakes (Stortjärnen) has littoral peatlands along roughly half of its shoreline, with peat virtually floating in the lake water,
potentially releasing DOC directly or via groundwater to the lake. This particular lake showed significant increases in DOC and $a_{420}$ with increasing WTT (see Table S1 statistics), possible due to direct inputs from the littoral peat. Moreover, since peatlands in an area close to the Stortjärnen lake have been shown to have particularly high DOC concentrations during low flow periods (Laudon et al., 2011), it is possible that Stortjärnen makes a special case where even small diffuse hydrological inputs from the peatlands surrounding the lake can be sufficient to raise the DOC concentration during low flow. Nonetheless,
while it is interesting to note that DOC accumulation can overcome degradation in some small individual unproductive lakes, results from this study are too limited to generalize such patterns.

**4.3 Organic carbon transformation processes**

To adequately understand our results it is necessary to bring the mechanisms of DOC decay in the lakes to attention. Previous studies in the region have shown that non-pigmented low molecular weight carbon (LMWC) is selectively used by bacteria in brown-water streams and lakes (Berggren et al., 2010b;Berggren et al., 2010a). In agreement with these previous results, the LMWC indicator $a_{254} : a_{365}$ (see relationship between LMWC and $a_{254} : a_{365}$ in Fig. S3) decreased significantly with WTT in brown-water lakes in this study, which together with increasing $a_{420} : DOC$ of the same lakes suggests that low-pigmented fractions were consumed more (and/or produced less) compared with colored bacterial substrates. Hypothetically, the microbial processing *per se* could have increased the pigmentation of DOC by the excretion of humic-like chromophoric molecules by bacteria (Shimotori et al., 2009;Tranvik, 1993;Guillemette and del Giorgio, 2012), although this appears less likely. Regardless, our results strongly suggest that microbial processes contributed to the increase in $a_{420} : DOC$ and the strong decrease in $a_{254} : a_{365}$ found coherently in both our dark biological decay experiment and in the brown-water lakes *in situ*. The fact that these lakes showed the same in situ optical changes over time as shown in the dark bioassays indicates that dark biological degradation processes dominated the DOC transformation in the brown-water lakes, probably because of high optical density which prevented photo-degradation from most of the water column.

For the clearest lakes, the prevailing mechanism behind the DOC transformation was obviously different than that in the brownest lakes, at least in the epilimnetic waters exposed to sunlight. The clear epilimnetic lake waters showed increases in $a_{254} : a_{365}$, which is the expected pattern for systems where UV light transformation dominates (Dahlén et al., 1996) and where LMWC is produced by photochemical processes (Bertilsson and Tranvik, 2000). In agreement, we also found that $a_{254} : a_{365}$ increased in the laboratory UV light experiment. Thus, the relative role that UV light processing played for the qualitative DOC transformations was much larger in clear lakes than in highly colored lakes. In absolute terms, however, some studies suggest that the absolute light-induced color loss is similar for brown and clear lakes, given equal incoming surface irradiation, even if this loss in color is distributed in different ways over the water column (Granéli et al., 1996;Koehler et al., 2014). In our study the mean absolute $a_{420}$ loss in the four clearest epilimnetic sites was 1.0 $m^{-1}$ $yr^{-1}$ (Fig. 2D). Based on Granéli et al. (1996) it can be speculated that the same processing happened also in the most colored of our lakes, but that this change was too small to be distinguished from the high background $a_{420}$ level of 11-13 $m^{-1}$ in these lakes.

Survey lakes of intermediate color generally showed no change in the DOC quality transformation indicators with WTT (Fig. S5). Rather the opposing influences on DOC quality by microbial and photochemical processes, respectively, appeared to have cancelled each other out. Given potential positive feedbacks between these two processes on the overall DOC mineralization, e.g. production of LMWC by photo-oxidation which in turn fuels microbial degradation (Bertilsson and Tranvik, 2000), it would the theoretically possible that total DOC losses were largest in lakes of intermediate color. Therefore, it might not be a coincidence that the only two lakes showing significant DOC losses in our study (Mångstrettjärn and Lapptjärn) had the intermediate mean color of 5.5-6.3 $m^{-1}$ (see Fig. 2c; Table S1).

**4.4 Comparison between temporal and spatial assessments**

In spatial comparisons of lakes (or lake sub-basins) with increasing theoretical residence times, the color and DOC often decrease in a predictive way (Köhler et al., 2013;Curtis and Schindler, 1997) and colored DOC is generally lost preferentially over non-colored DOC (Weyhenmeyer et al., 2012). However, Berggren et al. (2009) demonstrated in a temporal analysis that, contrary to the expectation, the specific color of DOC can increase during water transit time through headwater lakes. The present study presents an explanation to such observations, showing that color per unit of DOC can either increase or decrease temporally in different types of lakes, likely depending on the relative importance of microbial and UV light processing, respectively. Thus, while there is an existing view that DOC transformation processes in lakes across the landscape can be understood based on spatial comparisons (Curtis and Schindler, 1997;Weyhenmeyer et al., 2012), we show here that temporal analyses are crucially needed to draw correct conclusions about the change in DOC with WTT.

Nonetheless, comparing the mean $a_{420}$ and $a_{420}$ : DOC values with mean WTT of the different lakes (Table 1; Fig. 2d), it appears as if color would be rapidly and preferentially lost across all lakes combined. This brings up a paradox: If it is not WTT *per se* that is responsible for the patterns in color and DOC properties; why do short-turnover lakes systematically have higher color and $a_{420}$ : DOC compared with the long-turnover lakes? Here we are not able to come up with a general solution, but it should be stressed that our specific study lakes have catchments of differing properties and hydrological functioning, likely influencing the observed patterns. In particular, the five lakes with the shortest WTTs have relatively large catchment areas drained by streams, whereas the four lakes with the shortest WTTs lack permanent streams (see Table 1 and Text S1). This is important because lakes with inflowing streams are extensively connected to riparian soils that are key sources for DOC (Laudon et al., 2011), whereas lakes without inlet receive relatively more input from DOC-poor deeper ground water (Tiwari et al., 2014) that has low color per unit DOC (van Verseveld et al., 2008). Moreover, the five short-WTT lakes are located in flat lower parts of the catchments with large direct connections to peatland areas, while remaining lakes are surrounded by hilly forest. As peatlands release larger amounts of highly colored DOC than forests (Ågren et al., 2008b), high peatland connectivity plausibly contributed to the brown character of the short-WTT lakes. Thus, the lakes with long WTTs in this study may not be clear primarily because of their long water transit times *per se*, but because they are disconnected from the key sources of colored DOC in the catchment, i.e. peatlands and upland riparian soils.

**4.5 Conceptual implications**

Altogether our results suggest a conceptual framework for DOC transformation (Fig. 5) with distinctly different color trajectories under 'brown-water' and 'clear-water' regimes, respectively. In the brown-water regime biological processes dominate, leading to small changes in color over time. In a clear-water state, photochemical processes play a relatively much larger role, resulting in substantial decreases over time in both DOC concentrations and in color. From this proposed concept (Fig. 5), the circumstances which allow for development of a lake into a brown-water or clear-water system, respectively, can be discussed. Considering that boreal watercourses show at least two orders of magnitude spatial variation in color (Lapierre

et al., 2013;Temnerud et al., 2014), the color of the inflowing catchment runoff water is beyond doubt a key factor determining the regime. Additionally, it can be speculated on the potential for regime shifts to occur in lakes that are close to the border between the clear and brown-water regime, e.g. by rapidly increased input of unprocessed DOC from the catchment during extreme discharge episodes such as described by Raymond et al. (2016). However, tipping over a brown regime into a clear regime should be relatively difficult, given the inefficient color loss in brown headwater lakes (Fig. 5). This view is consistent with laboratory studies indicating that biological decay needs to proceed for a few years before substantial DOC and color exhaustion in brown water lakes (Koehler et al., 2012).

Our results do not contradict the findings from previous studies that have proposed selective loss of pigmented DOC in freshwater networks (Ilina et al., 2014;Weyhenmeyer et al., 2012;Köhler et al., 2013). Rather, this study can help explain why previous studies have not been able to detect the decreasing color with increasing water residence time in headwaters (Müller et al., 2013), where many of the lakes presumably follow the dynamics of the brown-water regime (see Fig. 5). The decreasing absorbance with increasing water residence time has mainly been possible to model at non-headwater sites with a relatively low color level (Müller et al., 2013;Weyhenmeyer et al., 2012). For such sites, decrease in color per unit DOC could be expected from photo-processing (Molot and Dillon, 1997), iron flocculation (Weyhenmeyer et al., 2014) or DOC replenishment along the aquatic continuum, e.g. from algal sources that selectively adds carbon of low degree of pigmentation (Creed et al., 2015).

## 5 Summary and conclusions

In summary, our results exemplify how individual brown-water lakes may not conform to the general reported pattern of selective removal of colored constituents in freshwaters, but rather show sustained level of pigmentation regardless of WTT variations. Thus change in WTT, e.g. due to a potentially wetter future climate, has no universal effect on lake color, at least not over the ranges in WTT that we studied. However, if combined with changes in the absorbance of catchment runoff water, an intensified hydrological cycle could possibly cause regime shifts in headwater lakes, where e.g. clear-water lakes renewed with more colored water relatively quickly will transform into brown-water systems. Conceptually, our study challenges the view of the aquatic network as a single continuum of DOC processing. In headwaters, the functioning of different aquatic networks depend on which DOC transformation process that dominates.

## Competing interests

The authors have no conflict of interest to declare.

## Acknowledgements

We thank Ann-Kristin Bergström and Anders Jonsson for valuable discussions and help with field logistics. Anna Nilsson assisted in laboratory nutrient addition tests on lake Nedre Björntjärnen. We also thank Marcin Jackowicz-Korczynski for

general laboratory support. The following assistants contributed in the field: E. Geibrink, I.-M. Blåhed, J. Gustafsson, K. Heuchel, J. Johansson, S. Prideaux, M. Myrstener, A. Aguilar, L. Lundgren, W. Lidberg, D. Karlsson, T. Andersson (deceased) and N. Lindqvist. M.B. was supported by "Multistressor" - a strong research environment funded by the Swedish research council FORMAS (grant #217-2010-126). The lake monitoring was financed by a FORMAS grant to J.K. (grant #210-2012-1461), with some additional funding from SITES (www.fieldsites.se).

**Available supplementary information** contains extended methods (Text S1), a table (S1) and five figures (Figures S1-S5).

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

Table 1. **Characteristics and sampling details for the nine study lakes.** Variables from left to right: decimal degrees (DD) WGS84 latitude (Lat) and longitude (Lon) coordinates, water transit time (WTT), area of lake ($A_{lake}$), mean depth of lake ($Z_{mean}$), area of catchment ($A_{catchment}$), ratio between the absorbance at the wavelengths of 254 nm and 365 nm ($a_{254} : a_{365}$), carbon specific absorbance at 420 nm ($a_{420} : DOC$), dissolved organic carbon (DOC), absorbance at 420 nm ($a_{420}$), number of epilimnetic and hypolimnetic samples (n), and study years. The WTT and chemical characteristics are shown as mean epilimnetic values across all sampling dates.

| Site name | Lat (DD) | Lon (DD) | WTT (yrs) | $A_{lake}$ (km$^2$) | $Z_{mean}$ | $A_{catchment}$ (km$^2$) | $a_{254} : a_{365}$ | $a_{420} : DOC$ | DOC (mg C L$^{-1}$) | $a_{420}$ (m$^{-1}$) | n (epi/hypo) | Years (20XX) |
|---|---|---|---|---|---|---|---|---|---|---|---|---|
| Fisklösan* | 64.150 | 18.800 | 0.94 | 0.017 | 2.1 | 0.089 | 4.55 | 0.32 | 7.7 | 2.5 | 38/28 | 11-14 |
| Nästjärnen | 64.160 | 18.777 | 3.13 | 0.010 | 4.2 | 0.034 | 4.50 | 0.35 | 7.6 | 2.7 | 40/30 | 11-14 |
| Mångstrettjärn | 64.251 | 18.762 | 1.44 | 0.018 | 5.3 | 0.141 | 4.12 | 0.48 | 11.5 | 5.5 | 41/31 | 11-14 |
| Lapptjärn* | 64.237 | 18.790 | 0.67 | 0.020 | 2.5 | 0.168 | 4.12 | 0.48 | 13.1 | 6.3 | 41/31 | 11-14 |
| Lillsjöliden[‡] | 63.845 | 18.616 | 0.19 | 0.008 | 3.8 | 0.254 | 4.10 | 0.49 | 15.7 | 7.6 | 35/33 | 12-14 |
| Nedre Björntjärnen*,[#] | 64.122 | 18.785 | 0.30 | 0.032 | 6.0 | 3.249 | 4.02 | 0.57 | 18.9 | 10.7 | 59/29 | 06-07, 11-14 |
| Struptjärnen[‡] | 64.023 | 19.489 | 0.34 | 0.031 | 3.8 | 0.791 | 4.03 | 0.55 | 21.4 | 11.7 | 38/29 | 12-14 |
| Övre Björntjärnen[#,‡] | 64.123 | 18.779 | 0.19 | 0.048 | 4.0 | 2.840 | 4.04 | 0.55 | 21.9 | 11.9 | 72/45 | 06-07, 09, 11-14 |
| Stortjärnen | 64.261 | 19.763 | 0.41 | 0.039 | 2.7 | 0.817 | 3.81 | 0.63 | 20.9 | 13.3 | 31/29 | 12-14 |

*Elevated inorganic N concentrations 2012-2014, by 0.1 mg N L$^{-1}$.

[#]Lake included in the focal study of the Björntjärnarna catchment

[‡]Lake selected for laboratory incubation experiments

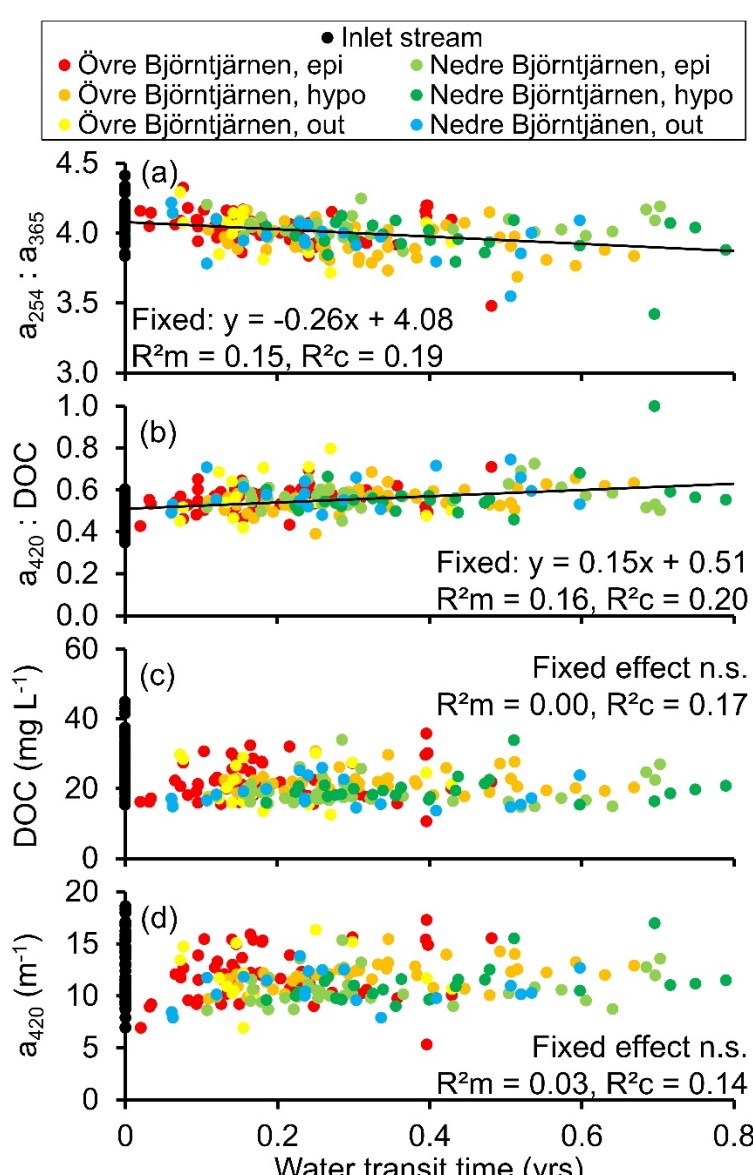

**Figure 1. Relationships between dissolved organic carbon (DOC) properties and the water transit time in two chain lakes with 92% shared catchment area**. The y-axis variables are (A) ratio between the absorbance at the wavelengths of 254 nm and 365 nm, (B) ratio between absorbance at 420 nm and DOC, (C) DOC concentration and (D) absorbance at 420 nm (n = 260, study years 2006-2014). Solid lines are based on significant (p < 0.01) fixed effects coefficients and intercepts. The $R^2$m shows marginal $R^2$ for fixed effects, where water transit time is the fixed effect, and the $R^2$c refers to conditional mixed effects models where site is included as a random effect (on intercept only, not slope). Inlet samples during drought (lower 5 percentiles of flow) are not included since drought inflow makes a negligible contribution to the water that resides in the lakes. In Fig. S4 all DOC property data from this figure is plotted over a calendar time line.

Abbreviations: n.s. – not significant; epi – epilimnion; hypo – hypolimnion; out – outlet.

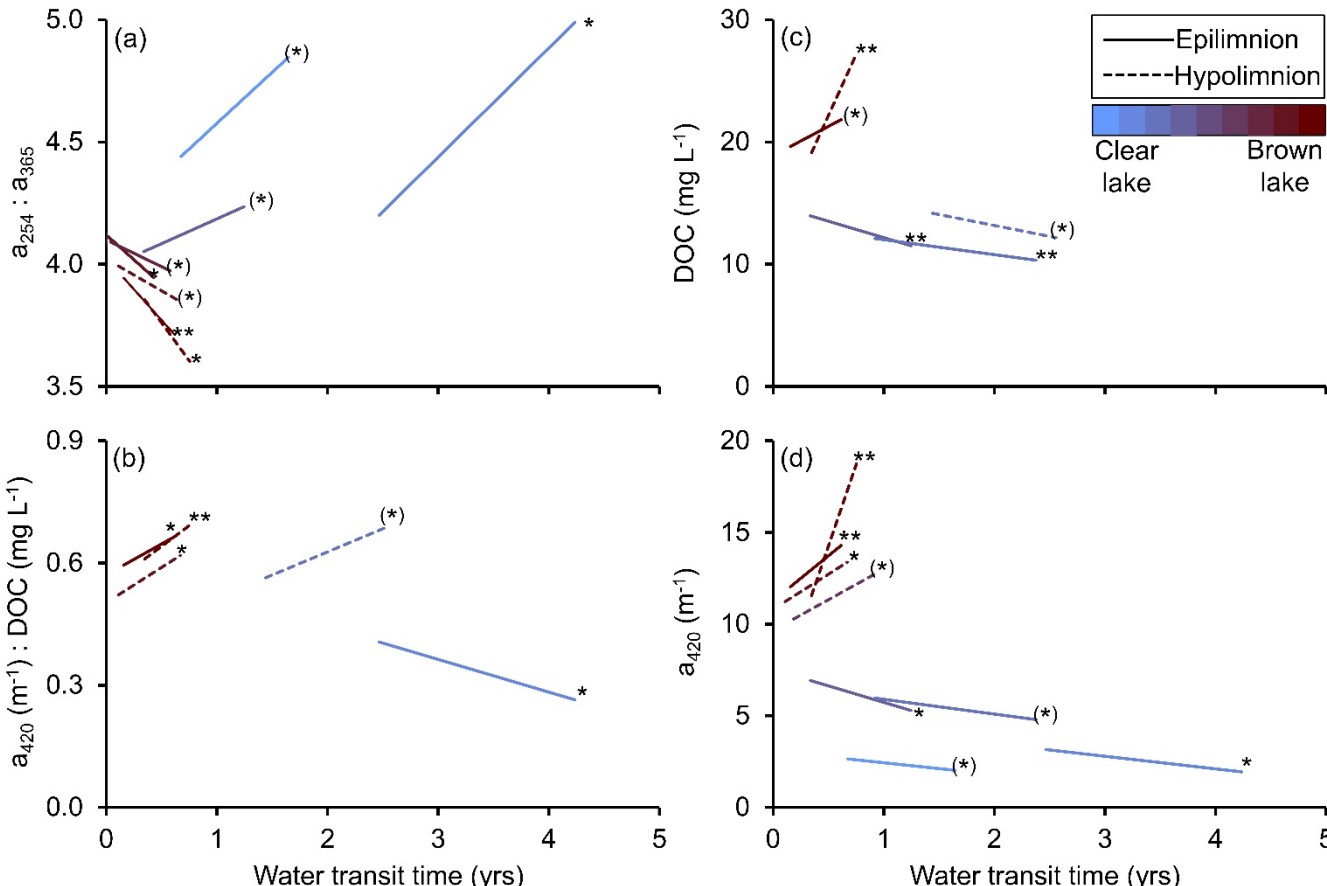

**Figure 2. Linear regression lines for significant[†] relationships between different properties of DOC and the water transit time, shown for individual epilimnia (solid lines) and hypolimnia (dashed lines) of nine lakes in northern Sweden**. Y-axis variables are: (A) the absorbance ratio $a_{254} : a_{365}$; (B) ratio between absorbance at 420 nm and DOC concentration; (C) DOC concentration and; (D) absorbance at 420 nm. Non-significant regression lines are not shown. For full statistical details and data plots, see Table S1 and Fig. S5, respectively.

[†]Significance: * $p < 0.01$; ** $p < 0.001$; (*) marginally significant $p < 0.05$

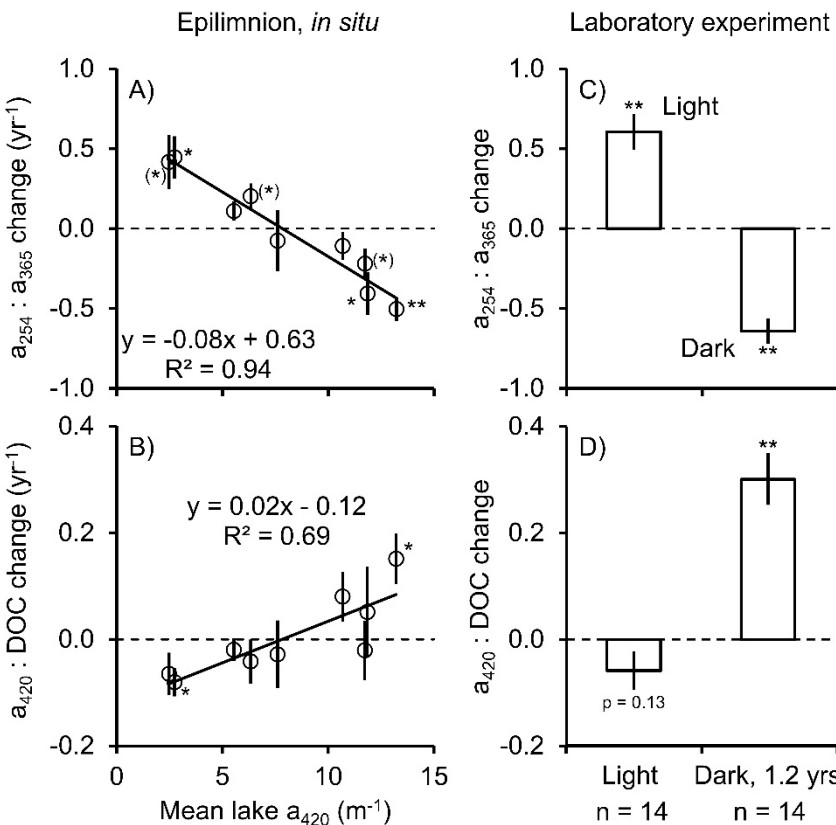

**Figure 3. Changes in DOC quality over time** as indicated by the optical indices $a_{254} : a_{365}$ (unitless) and $a_{420} : DOC$ ($m^{-1} : mg\ L^{-1}$), shown separately for (A-B) the lake epilimnia and (C-D) laboratory experiments. In panels A-B, the y-axis value of each symbol (± error bar) shows the linear slope (± standard error) of $a_{254} : a_{365}$ or $a_{420} : DOC$ as function of the water transit time in a certain lake[#]. Symbols with asterisks represent individual changes (*in situ* data regression slopes; see Fig. 2) that are significant[†]. The x-axis values show mean absorbance ($a_{420}$) of the lakes. Panels C-D show changes (stop minus start values; mean ± standard error) in $a_{254} : a_{365}$ and $a_{420} : DOC$ during light (~100 MJ absorbed per $m^3$ of water) and dark (1.2 yrs) experiments performed on water from three of the study lakes. Bars with asterisks show changes that are significant[†] (2-tailed paired t-test, comparing initial and ending conditions).

[#]See Table 1 for sample numbers and descriptions of each lake

[†]Significance: * $p < 0.01$; ** $p < 0.001$; [(*)] marginally significant $p < 0.05$

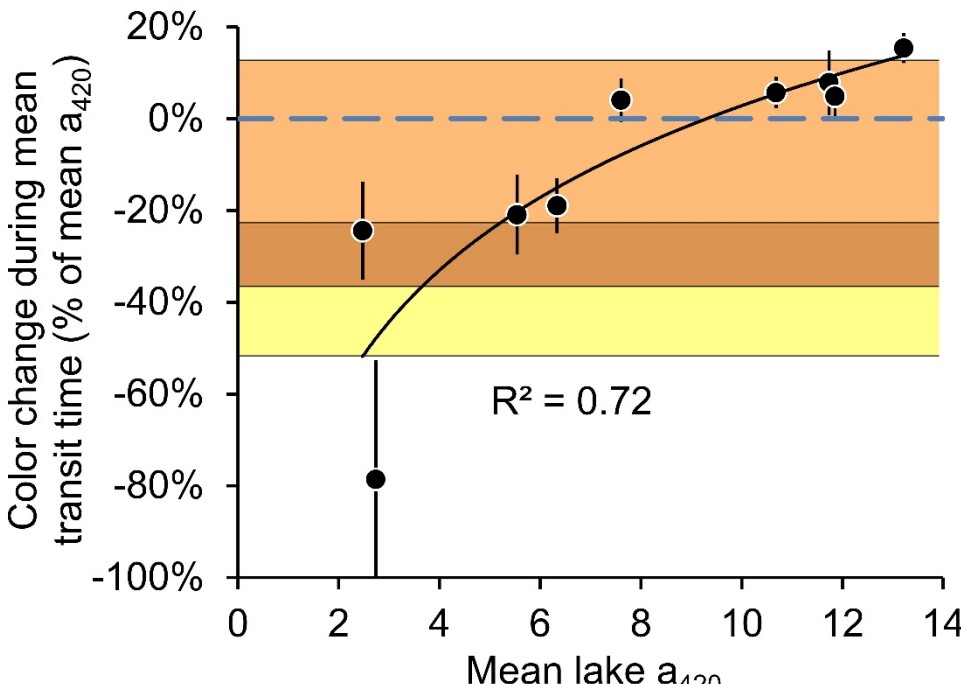

**Figure 4. Percent change in epilimnetic color ($a_{420}$; filled circles) during water transit through nine boreal lakes**, plotted over a gradient of increasing mean lake color (y = 0.39ln(x) - 0.87, p < 0.01). The change in color (± error bar) is based on the *in situ* data regression slopes (± SE) from Table S1, multiplied by the mean water transit time in each lake and expressed in proportion to lake mean $a_{420}$. The blue line denotes zero change. For significance of the individual color loss terms, see the original epilimnetic $a_{420}$ regression models in Table S1 or Fig. S5, where the lakes are aligned in the same order as here, from clear to brown. For a qualitative comparison, yellow and brown areas, respectively (overlapping in the dark brown area), indicate ranges in $a_{420}$ change from the beginning to the end of light (~100 MJ absorbed per $m^3$ of water) and dark (1.2 yrs incubation) experiments performed in the laboratory.

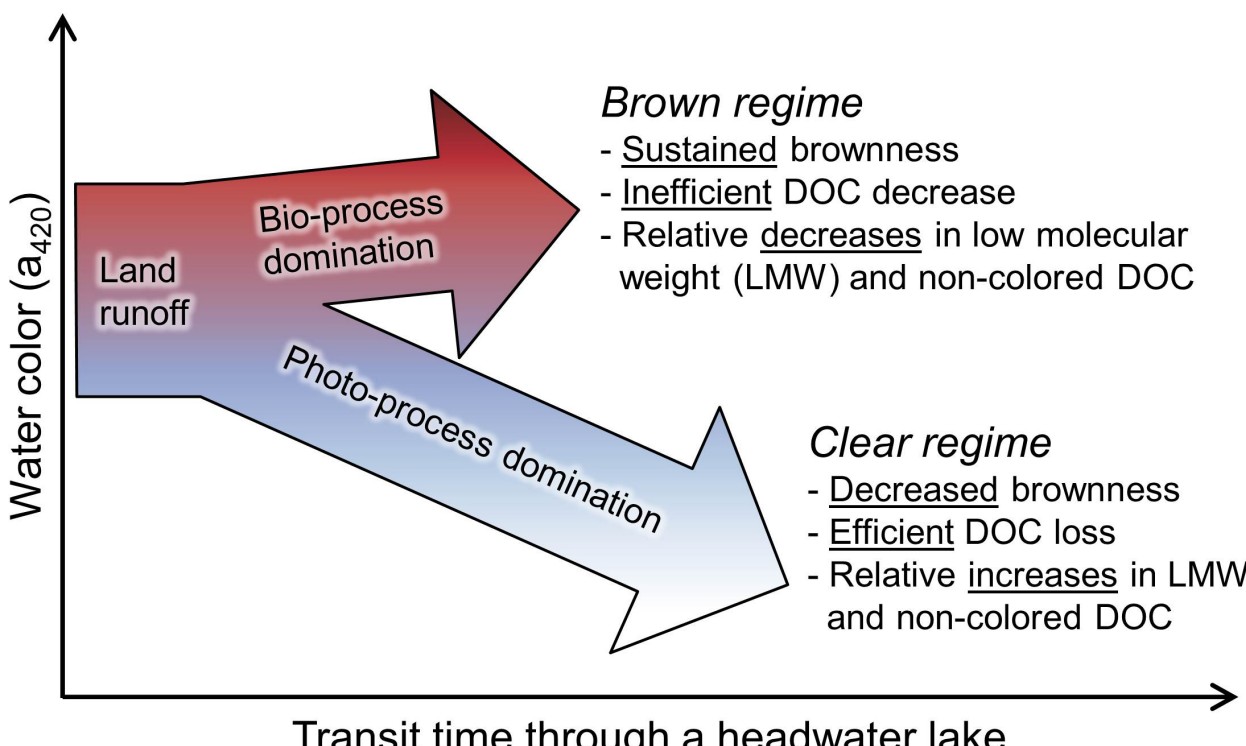

**Figure 5. Conceptual representation of the trajectory of color and dissolved organic carbon (DOC) characteristics in headwater lakes with increasing residence time under brown and clear lake regimes, respectively.** The figure is based on the observations from this study, where DOC processing in brown lakes is characterized by microbial consumption of low molecular weight (LMW) and non-colored DOC. Only the clear-water lakes show efficient reductions in color and DOC over time, facilitated by photochemical processing that remove colored DOC to a larger extent than microbial processes. Compare with original data from Figs. 2-4.