# Peer review of "Quality transformation of dissolved organic carbon during water transit through lakes: contrasting controls by photochemical and biological processes"

_Biogeosciences, 2017_

## Referee Comment (RC1) · Anonymous Referee #1 · 10 Aug 2017

**GENERAL COMMENTS**

In this paper, the authors test the hypothesis that coloured dissolved organic carbon (DOC) would be selectively lost in boreal lakes, following previous observations from large-scale studies. Based on field and lab data, they found that at the individual lake scale, DOC loss is largely dependent on ambient DOC color ($a_{420}$). They found that colour loss occurred in clear water lakes, whereas in brown water lakes DOC colour remained sustained over time.

[Figure]

These results have relevant implications for current debates about the role of lakes in carbon cycling and DOM processing, within the aquatic continuum as well as within the landscape, and therefore I strongly suggest this paper for publication. Findings of this paper are based on a complete data set that includes a large temporal period (7 years in Björntjärnarna catchment, as well as 3-4 years in 7 additional lakes) as well as a reasonable regional representativity of lakes with varying DOC and water transit time (WTT) conditions. The latter nicely showed how the "browning" level of lake water may be a main factor determining DOC reactivity within a lake. The authors argue that this factor may even overrule the effects of hydrology, even though I will partly question that below.

Unfortunately the authors did not explore the temporal perspective of their data set, as they pooled all the different sampled time points under a regression analysis approach. Showing some time series, even if it is in the supplementary material, would add completeness to the study; and may support some speculative paragraphs of the discussion, as I comment below.

Here follow some specific comments which I hope may help to improve the manuscript:

**SPECIFIC COMMENTS**

2.4 Water transit time assessments

P5, L 7: "The transit time, represented by the water that resides in a lake at a given moment, . . ." A time that is represented by a water volume sounds confusing. What about "The transit time of the water volume that resides in a lake at a given moment, . . ."
P5, L 9-10: It would help to add units of Voltotal, Flow rate and WTT.
This section describes the calculation of WTT values for epilimnion, hypolimnion, and inlet sites, but not for outlet, even though this data is later used in Fig. 1.

**2.5 Response variables**

P6, L 26-29: Is this a specific finding of this study, or from a previous study?

P6, L 23: "Besides DOC and $a_{420}$..." add mention that $a_{420}$ is an indicator for DOC color plus associated reference(s).

**2.6 Laboratory experiments**

This is the part of the paper I am less convinced of. These lab experiments, as standardised procedures, may be useful to compare the DOM reactivity from different sites/lakes, however I think their comparison with field data should be done with much care.

Dark experiments: I wonder how representative it is to incubate water for 15 months compared to what happens in the lake, where both the DOM and the bacterial community are continuously mixed with newly arrived molecules and cells. During these 15 months, did you check/control for nutrient limitation?

P7, L 26: "higher than in the dark control incubations": should it be without "control"? (no control was mentioned for the dark incubations).

P7, L 29: At the end, I suggest briefly mentioning that the measurements before and after the incubations were used to calculate the "change" in DOM properties (as it is later used in the results), and how this was calculated. May I point here that different units are presented in Figs 3 and 4.

**2.7 Statistics**

P8 L 5-7: Is there a reference to support this? Also, in order to evaluate the significance of the linear regressions, I strongly suggest the additional use of the $R^2$ (it is presented for the Björntjärnen lakes but not for the survey lakes regressions).

**3.1 Björntjärnarna chain lakes and Fig 1**

Fig 1. The colours between Inlet stream and Övre Björntjärnen epi are almost indistinguishable. However, here I wonder about the inlet and outlet sites. First I wonder if

it makes sense to add the inlet points to the analysis, since it is not affected by what happens in the lake. And about the outlet, I wonder to what WTT it is assigned to, since in the methods there is only a definition for the WTT of epi and hypo and inlet.

P18 L 3-4: This may not be needed, since the y-labels are already shown in the plots, and the variables described in the methods.

**3.3 Survey lakes and Figs 2 and 3**

In Fig. 2, the authors argue that there is a differentiated behaviour between brown-water lakes and clearer-water lakes. Even though this is later very neatly systematized in fig 3, I suggest adding this information somehow already in Fig 2, to help relate the plot with the description in the text. One suggestion would be to draw the lines in a colour indicating the corresponding DOC concentration, or $a_{420}$, as reported in Table 1. This would also allow seeing if any two lines of epilimnion and hypolimnion are paired.

On the other hand the two groups with opposite slopes, not only correspond to clear vs brown water lakes, but also they have very different ranges of variation of the WTT. For example, in Fig 2A, those lakes with negative slopes are also those with shortest ranges of variation of WTT. So here it is fair to wonder to what extent the slope is a statistical artefact resulting from the data not covering a similar range of variation. In fact, if all lakes were pooled together, the relationship between a254:a365 would be positive (we do not see the points in the graph, but I am joining the regression lines), and the same can be said for the other panels (if all points were to be pooled together, they would follow the trend described by those lakes with larger WTT ranges). With this I do not mean to invalidate the results, but maybe some more information could be added in order to emphasize the validity of these correlations, like adding the $R^2$, or plotting their corresponding data points to evidence a clear linearity. I think it is important to solidify these results, as later they lead to intriguing interpretations like DOC concentrations increasing with longer WTT.

An interesting result that can also be drawn from figs 2-3, is the fact that those lakes

with intermediate colour levels are less responsive to changes in WTT (slopes not significant). This would imply that this kind of lakes are less sensitive to hydrological variability and therefore less affected by hydrological events like rainfall or drought. This could also be mentioned/discussed in the text.

**3.4 Experiments**

P 9 L 25: "similar to the changes observed over time" according to the caption of figure 3, it is not a variation over time but as a function of WTT.

**3.5 Overall color loss**

P9 L 29-30: "we multiplied the in situ rate of epilimnetic color loss in the survey lakes (same as slopes in Fig 2d) with the mean water transit time for the respective sites to find out how much total change there was in water color upon transit": I do not understand how this becomes a percentage of color loss, I suggest that this is more explicitly stated.
Then, this "relative color change" is compared with the percentage change in the experiments. The calculation of the latter is never explained in the text, I suggest briefly explaining that.

**Discussion**

P10 L16-21: I think it should be taken into consideration here that the brownest lakes also had much shorter ranges of variation of WTT.
P10 L 23-31 With the data set you have, including 3-7 year time series, you would not need to speculate about that. Why not just check how $a_{420}$ and DOC change over time, or seasonally, in the inlet and outlet of the Björntjärnen lakes?

**Conclusions**

P13 L 2 "brown headwater lakes": or just "brown-water lakes"?
P 13 L 4 "Thus change in WTT, e.g. due to a potentially wetter future climate, has no

universal effect on lake color": This is a too hard statement, considering that your data for brown water lakes only covered a small range of WTT values.

**TECHNICAL COMMENTS**

Some suggestions, even though I am not a native English speaker:

P2, L 26: "... as a result in temporal ...": as a result of temporal
P3, L 1: "selective" instead of "selected".
P3, L 6: "... and analysed using linear mixed effects regression" probably not necessary to be mentioned at this stage. If it is mentioned, though, it should be stated what the mixed effects regression was for.
P5, L 8: "that passes by" not necessary.
P7, L 30: "The response of" instead of "The response in"
P7, L 3-4: "goes up" and "goes down": may I suggest avoiding these. Maybe that could be replaced simply for "If this ratio increases with WTT..." and "but if $a_{420}$:DOC decreases...".

---

## Referee Comment (RC2) · Anonymous Referee #2 · 17 Aug 2017

**GENERAL COMMENTS**

In this work, the authors aim to determine the relevance of bio-and photo-degradation processes during the water transit time in individual lakes. The authors hypothesize that each process will prevail as a function of the color of the DOC compounds, so that biodegradation will target non-colored DOC while photo-degradation, colored DOC compounds. Using a complex data set at different temporal and spatial scales and including both field and experimental data, the authors found brown-water lakes to be dominated by biodegradation processes (not photo-degradation), which leads to their persistent brown-water color.

The authors present these results as contrasting with the current paradigm of loss of colored constituents of DOC along the inland waters continuum. However, they do not provide such a continuum (i.e. accumulated water residence time along the landscape), they do not evaluate the molecular composition of DOC and, the presented here are net changes (i.e. including production and degradation of DOC) but they are not discussed as so. I consider the partitioning between photo and bio-degradation processes a key question to complete our knowledge on the pathways of C processing in inland waters. But because of this relevance, I ponder indispensable that the authors clarify those concerns above and the ones specified below (such as properly assessing the role of hydrology, improving the characterization of DOC or providing the complete results -the last specially affecting Figure 2-) before this manuscript can be considered for publication. I hope these comments are helpful and constructive.

**SPECIFIC COMMENTS**

**Abstract**

P1 L17: "photo-chemistry qualitatively dominated"…what does qualitatively mean here? That the changes in DOC quality were dominated by photo-decay? That you assess that in a qualitative (i.e. non-quantitative) way? Clarify in the text.
Also, photo-chemistry dominated the DOC or the CDOM transformation in headwater lakes? How is the production of non-colored DOC evaluated? Clarify in the text.

P1 L19: Was there a systematic relationship between color loss and WTT in clear-water lakes? Add this information also.

**Introduction**

P2 L17: Maybe biodegradation processes do not affect colored DOC preferentially, but that they do affect it at all has a stronger impact on the inland waters C budget than the consumption of in-situ produced DOC. Add information on the DOC sources and their relevance on the C budget here.

P2 L22: Available references on "efficient" photo processing, showing how polyphenolic, aromatic compounds are mostly affected by photo reactivity (assessed at a molecular level) in black and boreal waters, are missing (e.g. Stubbins et al. 2010 L&O, Kellerman et al. 2014 Nat. Comm. and references therein).

P2 L27: I agree with the authors that the assessment of the variability of WTT within systems is very relevant. However, without assessing how that variability is linked to changes in color of runoff DOC, it is hard to attribute the changes in the lake just to in-situ biogeochemical processing. Clarify that here and incorporate that perspective throughout the text -see comments below-.

**Methods**

P3 L16: modify this sentence into "lakes are located in the boreal region, where nutrients" and provide a reference of that distribution.

P3 L30-32: Although, low effects of pH on the optical properties of DOM have been reported at the most frequent inland water's pH range (i.e. 5.5-7.5), they can be important at lower pH values (< 4.5), such as the ones included in this study. Accordingly, add a paragraph in the discussion stating which lakes presented these low pH values (i.e. 3.4) and how could that affect your absorbance measurements (some useful literature: Pullin and Cabanis et al., 2003, Geochim. Cosmochim Act.; Patel-Sorrentino et al., 2002 Wat. Res.; , Spencer et al., 2007, Wat. Res.).

P4 L6: Consider reporting Catchment area/ Lake area ratio as a more relevant variable to discuss epilimnetic WTT than catchment area alone.

P5 L3: Why using only 3 wavelengths if the whole spectra were available? Given the aim of the study, much more robust conclusions could be reached if other widespread descriptors such as SUVA254 and slope analysis were included, and I recommend their inclusion. Those descriptors are widespread, and in particular, spectral slope analysis, is recognized to provide further insight into DOM composition than absorption coefficients alone (see Helms et al. 2009 L&O, Loiselle et al. 2009 L&O). Package "cdom" in R could be a useful tool to perform that exploration.

P5: Calculations for outflow are nor provided but they are presented in Figure 1. Add this information here.

P6 L17: Are all the other catchments spatially independent? Even if the inlet streams are considered negligible, what about the accumulated time in the catchment (*sensu* Müller et al. 2013 Aq. Sci.)?

P6 L27: The relative contribution of LMWC to total DOC (%) should be used instead of the total concentration of organic acids. A higher total sum of organic acids could be just due to a higher DOC concentration. Thus, to clarify if samples have a higher relative contribution of LMWC compounds or just higher DOC, the relative contribution of LMWC to total DOC (%) should be used, and ideally both (LMWC for each sample and in % and in mgC L-1) shown in the Supplementary Information.

Also, is the correlation between a254:a365 and the organic acids positive or negative? Should be stated.

P7 L13: Bacteria might dominate the biomass, but still be predated by heterotrophic flagellates. How does the bacterial abundance looked during the experiments?

Moreover, 450 days is a very long period, which effects would have both the predation and the death of the bacterial community and subsequent mineralization of that biomass on the results? How fair is it to consider that these results reproduce the biodegradation process occurring in the field where lakes behave like chemostats not like batch incubations? Justify in the text, and discuss later the implications and assumptions that have to be done to compare both results in the discussion.

P7 L18: I agree that microbial processing can happen in the entire water column, but I believe the simultaneous action of UV and biodegradation cannot be discarded. On the one side and mainly, because photo-mineralization rates are faster than biodegradation rates. On the other side, because there are several situations where the entering water will be exposed to both ( i) water in the hypolimnion, would have been initially exposed to both UV and microbes when entering the lake, ii) under ice conditions, microbial activity would also be minimal due to low water temperature iii) during the ice-free period and at that latitude, daylight is almost for 24h). Thus, both processes are likely to occur also simultaneously or following the inverse sequence (photodeg --> biodeg). Justify that, considering the number of papers using the opposite approach.

The authors could also perform a much deeper exploration of the changes between layers with the temporal data available and in light of the results shown in Fig.2 on that direction.

P7 L25: Similarly for photo-decay than for bio-decay: even if a radiation equivalent to two years was applied, there was no water renewal considered. Discuss how well you expect this results to reproduce the process in the field.

P8L8: Where the assumptions fulfilled?

P8L11: Specify which variables are set as the fixed effects and as the random effects here.

**Results**

P8 L24: Is "the most dynamic lake" also the smaller lake (volume)? The one with bigger catchment? I missed that in the discussion later and to discuss the controls on the trends on WTT and color in the epi- and hypolimnion.

P9 paragraph 3.4: There are no details provided on what is considered "change" in the incubations. Also, changes in DOC and ideally DOC decay rate should be shown in Fig. 3

P9 L30: Provide details (e.g. units) of this calculation. Also, only the ones in Fig. 2 were included, or all the sites? Clarify. Also, looking at these figures, how does the reader know which are the "clearest" and "darkest" lakes? different symbols should be used. Moreover, that categorization should be clearly defined and the cut-off between both justified previously and based on values previously reported in the literature. Also, in Table 1, it should be an additional categorical variable stating if a lake is "clear" or "brown".

**Discussion**

P10 L10: Which impact could it have that WTT does not span a whole hydrological year? Discuss here.

P10 L13: "the quantitative photo-bleaching in the Björntjärna catchment", what do the authors mean? Was there a quantitative evaluation of that? What is the total DOC photo-bleached in the catchment? Also were those studies (Lindell et al. 2000; Vachon et al. 2016) using a similar approach?

P10 L17: If I am correct, now comes the only available definition of "brown" lakes. Also...what other variables define a brown or clear- water lake??

Could the authors relate these categories with e.g. morphological variables? (e.g. volume, catchment/lake area, peatland presence, etc). It feels somehow poor to discuss the change in color using a categorical variable built upon that same parameter. I recommend to provide a full multi-parametrical characterization of the two groups.

P10 L20: Müller et al. 2013 evaluated the influence of lateral water inputs. Could later inputs explain the patterns found here? Was there some assessment of lateral fluxes in the systems (e.g. groundwater inputs) so as to discard that from happening in some of the other brown-water lakes?? Discuss in the text.

P10 L30: How is it in Fig. S1b evaluated the contribution of runoff to total water and DOC? The authors do not explicitly evaluate this and they should do so. According to that figure, as runoff increased, WTT decreased. Therefore, we could expect the exported water/DOC during episodic flows to be flushed away from lakes also. As WTT turns longer after the flow, the DOC sources and thus composition, should also recover. To avoid that interpretation, the authors should explicitly evaluate the contribution of runoff to the budget, and discuss more in depth differences found in that sense between the different type of lakes (i.e. above and below one hydrological year, clear and brown) and their layers (epi *vs.* hypolimnion).

P11 L13: I consider the authors cannot conclude this, as there cannot be confident on the evaluation of the inputs performed, and that should be discussed at that point. Thus, "DOC accumulation can overcome degradation even in some small individual unproductive lakes" and it can be due to reduced degradation or to lateral terrestrial inputs. Add that discussion.

P11 L17: The authors should evaluate these processes always as a net result of production vs consumption. Thus, in brown-water lakes, the apparent decrease in LMWC is due to consumption above production. Opposite would hold true for clear-water lakes. Implications of acknowledging that are apparent and results need to be discussed under that light.

P12 L1: Thus, the total color loss might be the same in both type of lakes, but the relative loss in brown water much lower. So… if the brown water lakes correspond to the headwater and lower WTT lakes, terrestrial inputs being more important and frequent (lower WTT), could that color loss in brown lakes (even if just representing a small fraction of the total color) be indeed more important at the landscape level?

Discuss, and as previously stated, provide a better characterization (including morphology and relation with the catchment, especially with terrestrial inputs) of the two lake types (clear vs brown).

P12 L20: What does it mean that it eventually "takes over"? Which mechanism could then explain it? Are there no other environmental or morphological factors that can explain that? Which could be the temporal threshold and could that be related with the hydrology? Include these questions in the discussion.

P11 L23: I believe it is very bold to interpret the incubation results that way. They give us an idea of the changes caused by one mechanism, but they do not exclude other mechanisms to happen. All the potential processes that could produce these changes in in-situ lake CDOM should be discussed.

**Summary and conclusions**

The first sentence sounds contradictory. If only headwater lakes are being evaluated, then, it cannot be assessed a general freshwaters pattern.

I believe the fact that headwater streams present "a sustained level of pigmentation regardless of WTT variations" is extremely interesting, and the relationship of that with hydrology and input sources deserves a much deeper exploration, and I encourage the authors to move towards that direction. Otherwise, the affirmation that "the results may not conform to the general reported pattern of selective removal of colored constituents" without providing an evaluation of the DOC sources variability, does not hold firmly.

**Tables and figures**

**Table 1:** Provide volume or depth information. Provide the categorical variable: clear or brown.

**Figure 1:** use different symbol for inlet or black color, it cannot be distinguished. Also, add definition of the outlet calculation in methods. Without that information… Shouldn't "out" WTT be longer than "epi" WTT? Answer and clarify in the text.

**Figure 2:** I recommend fully re-working this figure and splitting it in two if needed. Above all, all data should be provided, for all lakes and layers, significant or not, so that the relationships not shown here could be evaluated by the reader. Moreover:

- The reader should be able to identify the lakes, to assess if the trends in the two layers are opposed or similar in each system.
- Also, it is impossible to assess the adequacy of the fittings without the points even if p-value is reported, and that is very important information.
- It is not clear which are the clear and which the brown water lakes, include that information in the legend.
- There seems to be two groups also as a function of WTT, how does that influence the results? e.g. in Fig 2d, where epilimnion and hypolimnion present completely opposite trends for the two age groups.
- Consider providing a summary table with the results of all the regressions, so the reader realizes how many fittings and which were not significant also.

**Figure 4.** It is not clear how that % is calculated (see previous comment). Also, are these changes significantly different from zero? Add that information as well as a zero-line. Clarify also in the caption that the slopes correspond to the ones in Fig. 2d. The reader should be able to identify to which line in Fig. 2d corresponds each dot in Fig. 4, modify accordingly.

**Figure 5:** The presence and contents of this figure should be re-evaluated once the suggested changes have been taken into account. Also, as it reads now, it is a bit like the chicken or the egg dilemma: are brown regime lakes brown because they have high water color? Or do they have color because of their brown regime? In other words, what is the progress on defining color regime only based on color?

**TECHNICAL COMMENTS**

P1 L13: "DOC quality and color"…if color and quality are considered separately, which variables are being used to describe quality besides absorbance? Isn't color quality of DOC? I suggest modifying into "changes in DOC color", as it most accurately describes the approach used here.

P1 L17: "Photo-chemistry" includes all the chemical effects of light, so that is not incorrect, but, as a "dominant process in DOC transformation in the epilimnia", do the authors specifically mean "photo-decay" or "photo-degradation?

P1 L20: Would "moreover" be more appropriate than "instead"?

P2 L2: Consider changing "and to cause" into "and cause"

P3 L1: Consider changing "selected" into "selective"

P3 L28: absorbance or absorption coefficient?

P6 L27: Fig. A2 should be Fig. S2?

P7 L29: "was" should be "were"

---

## Author Response (AR1)

Response to GENERAL COMMENTS

1. *In this paper, the authors test the hypothesis that coloured dissolved organic carbon (DOC) would be selectively lost in boreal lakes, following previous observations from large-scale studies. Based on field and lab data, they found that at the individual lake scale, DOC loss is largely dependent on ambient DOC color (a420). They found that colour loss occurred in clear water lakes, whereas in brown water lakes DOC colour remained sustained over time.*

   *These results have relevant implications for current debates about the role of lakes in carbon cycling and DOM processing, within the aquatic continuum as well as within the landscape, and therefore I strongly suggest this paper for publication. Findings of this paper are based on a complete data set that includes a large temporal period (7 years in Björntjärnarna catchment, as well as 3-4 years in 7 additional lakes) as well as a reasonable regional representativity of lakes with varying DOC and water transit time (WTT) conditions. The latter nicely showed how the "browning" level of lake water may be a main factor determining DOC reactivity within a lake. The authors argue that this factor may even overrule the effects of hydrology, even though I will partly question that below.*

   *Unfortunately the authors did not explore the temporal perspective of their data set, as they pooled all the different sampled time points under a regression analysis approach. Showing some time series, even if it is in the supplementary material, would add completeness to the study; and may support some speculative paragraphs of the discussion, as I comment below.*

Reply: We thank Reviewer #1 the thoughtful comments that have helped us to improve the manuscript. On the more specific remark about showing time series, we agree that showing raw data for the response variables over calendar time is a good idea. In the revised ms we will add such time series to the supplementary material, at least for the Björntjärnarna catchment, and we will use this material to support the discussion regarding e.g. seasonal timing of the DOC export (see also reply to specific point #20 below).

**Author's change: A new figure (present Fig. S4) has been added, plotting all raw data for DOC and its characteristics (same data as used for Fig. 1) over Gregorian calendar time. See specific point #20 below regarding related changes in the discussion.**

Response to SPECIFIC COMMENTS

1. *2.4 Water transit time assessments P5, L 7: "The transit time, represented by the water that resides in a lake at a given moment, : : :" A time that is represented by a water volume sounds confusing. What about "The transit time of the water volume that resides in a lake at a given moment, : : :"*

Reply: Changed as suggested

**Author's change: Implementation of the suggested change found on p. 5, l. 20**

2. *P5, L 9-10: It would help to add units of Voltotal, Flow rate and WTT.*

Reply: As long as the right type of physical quantities are entered (e.g., 'volume', 'time'), the input units of preference do in principle not matter – what goes in is what comes out. However, looking closer at our manuscript we noted that the physical quantity 'flow rate' was not well defined. Therefore, we now define this property as 'volume per unit time'.

**Author's change: Definition of flow rate as 'volume per unit time' added to p. 5, l. 22**

3. *This section describes the calculation of WTT values for epilimnion, hypolimnion, and inlet sites, but not for outlet, even though this data is later used in Fig. 1.*

Reply: In the revision we will clarify that complete mixing of the epilimnion is assumed, such that outlet water is equal in its properties (including WTT – time spent in lake) to epilimnetic water.

**Author's change: Explanation to outlet WTT assumptions added on p. 6, l. 30 – p. 7, l. 2**

4. *2.5 Response variables P6, L 26-29: Is this a specific finding of this study, or from a previous study?*

Reply (original): Both! Our results confirmed the expectation based on a handful of previous studies, among them Panneer Selvam et al (2016, JGR Biogeosci 121:829-840) and Lapierre et al (2013, Nature Comm 4: 2972). We will add one or two references to support this expectation/finding.

Reply (new): By mistake, we first looked at 'page 7 line 26-29' instead of 'page 6 line 26-29'. Thus when originally replying, we thought that the Reviewer referred to the statement on page 7: '*bacterial DOC processing during irradiation was considered negligible…*'. However, what the Reviewer actually asked about was the correlation between a254 : a365 and the concentrations of low molecular weight DOC, mentioned on page 6. Nonetheless, the answer to the Reviewer is the same, i.e.: '*Both! Our results confirmed the expectation based on a handful of previous studies*'. In other words, the correlation in question is not first shown in this study, but rather something known. We brought in the data (now found in Fig. S3) to validate the assumed indicator value of a254 : a365, supporting the methodological choice of using a254 : a365 as a DOC quality variable.

**Author's change: We have changed the phrasing to clarify that '*In agreement with the expectations based on Berggren et al. (2010a), $a_{254} : a_{365}$ in this study was positively correlated to…*' (p. 7, l. 17-18)**

5. *P6, L 23: "Besides DOC and a420: : :" add mention that a420 is an indicator for DOC color plus associated reference(s).*

Reply: Changed as suggested

**Author's change: Phrase changed to '*the color indicator $a_{420}$ (Weyhenmeyer et al., 2012)*' (p. 7, l. 9-10)**

6. *2.6 Laboratory experiments This is the part of the paper I am less convinced of. These lab experiments, as standardised procedures, may be useful to compare the DOM reactivity from different sites/lakes, however I think their comparison with field data should be done with much care.*

   *Dark experiments: I wonder how representative it is to incubate water for 15 months compared to what happens in the lake, where both the DOM and the bacterial community are continuously mixed with newly arrived molecules and cells. During these 15 months, did you check/control for nutrient limitation?*

Reply: We agree with the reviewer that the laboratory incubations do not represent exactly what happens in the lakes *in situ*. This comment helped us see that the purpose of our laboratory experiments was not sufficiently well described in the original submission. Briefly, what we wanted to achieve was experimental conditions during which either 1) photochemical reactions strongly and dominantly influenced the DOM transformation, or 2) microbial degradation strongly dominated the DOM transformation. Thus the experiments were designed such that the response to a large light dose or a long microbial process time in the dark was measured. While we don't believe that such experiments mimic lake *in situ* conditions in an adequate way, they do provide qualitative information about how the DOM responds to the isolated effects of photochemical and biological decay. Interestingly, the patterns of DOM transformation found in dark experiments well matched the *in situ* DOM quality changes observed in dark (brown or hypolimnetic) environments, while our light experiments matched the qualitative patterns in DOM transformation in clearer and more light-exposed environments *in situ*. These findings are supporting our interpretations.

Regarding nutrient limitation, Jansson et al (2001, Freshw Biol 46:653-666) showed that the bacterial metabolism in lakes of our study area (including Björntjärnarna) was decreasingly dependent upon inorganic nutrients with increasing DOC concentrations. This may seem counter-intuitive but agrees with the results by Soares et al (2017, Biogeosci 14, 1527-1539), showing that the DOM in these lakes includes large amounts of bioavailable DON and P while the humic DOC is relatively more difficult for microbes to degrade. Relatively high P bioavailability in streams of the region has also been shown by Jansson et al (2012, L&O 57:1161-1170). Thus the higher the DOC, the less likelihood of nutrient limitation. In the laboratory experiments, the DOC was ca 15-20 mg C/L, which is a range representing conditions when nutrients are not expected to limit the bacterial metabolism in lakes of the study area (Jansson et al, 2001).

Therefore, based on the above, in the revised manuscript we will provide a clearer rationale for the experimental design of our study. We will also highlight that the experimental results only provide qualitative information about how the DOM responds to different types of decay – it is not possible to make quantitative comparisons. Finally, we will explain why there are good reasons to expect that there was no overriding nutrient limitation in our dark incubations.

**Author's change: The following text has been inserted to clarify that '***We performed laboratory experiments on water from three catchments to disentangle the isolated effect on DOC quality by UV light degradation from that of microbial processing. The purpose with the experimental design was to create conditions during which either 1) photochemical reactions strongly and dominantly influenced the DOC transformation, or 2) microbial degradation strongly dominated the DOC transformation. Therefore, we measured the DOC quality responses to a large light dose or a long microbial process time in the dark. However, the experiments were not designed to reflect in situ decay rates.***' (p. 7, l. 29 – p. 8, l. 1)**

**We have also added the following text regarding potential nutrient limitation: '***We assumed that there was no nitrogen or phosphorus limitation due to the high concentrations of DOC (~15 mg L-1), known to be associated with high organic nutrient bioavailability in lakes (Soares et al., 2017) and streams (Jansson et al., 2012) of the region. Our assumption is supported by Jansson et al. (2001), who showed that bacterial metabolism in humic lakes is generally not nutrient limited when DOC is higher than ~15 mg L$^{-1}$***' (p. 8, l. 10-14)**

7. *P7, L 26: "higher than in the dark control incubations": should it be without "control"? (no control was mentioned for the dark incubations).*

Reply: Changed as requested

**Author's change: Word 'control' removed (p. 8, l. 27)**

8. *P7, L 29: At the end, I suggest briefly mentioning that the measurements before and after the incubations were used to calculate the "change" in DOM properties (as it is later used in the results), and how this was calculated. May I point here that different units are presented in Figs 3 and 4.*

Reply: We agree that these things need better explanation. Indeed what we calculate and present in e.g. Fig 3b is the before→after incubation difference in DOM properties. The Reviewer is also correct that we present data from the same laboratory incubations using a separate unit in Fig 4. In the case of Fig. 4 the relative (%) change in color from the beginning to the end of the incubation is shown (as shaded areas). In contrast Fig. 3b shows the absolute changes in DOM properties from beginning to end of the incubations. In the revision we will clarify and explain how the different variables were calculated from the laboratory incubation data.

In the revision clarifications/explanations with regard to the above will be implemented both in the methods section and in the results section where the data is presented, e.g. in Figs 3-4 and their captions. Additionally, as explained in response to point #6 above, our revised manuscript will be clearer about the fact that we only mean to compare experimental data and field data in a qualitative way. Thus in Figs. 3-4 the point is not to show absolute agreements in the rates of DOM property change between field and laboratory measurements, respectively. We rather mean to demonstrate patterns of agreements in the directions and relative magnitudes of the changes.

**Author's change: The following text was added to p. 8, l. 30-34: '*The DOC and absorption coefficients were analyzed before and after each experiment, and the changes from beginning to end of the incubation (final value minus start value) were calculated for the absorbance ratio $a_{254} : a_{365}$ and the carbon color indicator for $a_{420} : DOC$. In addition, the relative change (%) in color ($a_{420}$) from beginning to end of the incubations was calculated and used for a qualitative comparison with percentage loss in $a_{420}$ during water transit through the different lakes in situ*'. Additionally the Figure 3-4 captions now mentions how the change in rates during experiments was calculated from beginning (start) to final (stop) values.**

9. *2.7 Statistics P8 L 5-7: Is there a reference to support this? Also, in order to evaluate the significance of the linear regressions, I strongly suggest the additional use of the R2 (it is presented for the Björntjärnen lakes but not for the survey lakes regressions).*

Reply: We base this reasoning on standard tables of critical values for the significance of correlations. A higher R2 is needed to get significance at the 0.01 level compared to the 0.05 level. A similarly higher R2 is needed to maintain significance if the number of observations is cut to half. Therefore, changing the significance level from 0.05 to 0.01 is roughly equivalent to losing half of the independent observations. However, here we should emphasize that this is rough and not exact. In the revision, we will explain better why we consider the alpha scaling to roughly (i.e., not exactly but fairly close) compensate for the temporal autocorrelation. We believe that this alpha adjustment is the simplest and most straightforward way to avoid granting significance too generously.

There are more advanced and perhaps mathematically/statistically correct ways to correct for temporal autocorrelation (e.g., based on bootstrapping), but then more complicated statistical procedures would have to be added. If (and only if) the Reviewers/Editor think it is worth spending the extra manuscript space on advanced correction procedures for temporal autocorrelation, then we would follow the recommendation and add this. It would not change the results or conclusions in our manuscript in any important way.

We could also explicitly add an autocorrelation term to the mixed effects modelling, but this this could only be applied to the Björntjärnarna catchment data where we use LMER, i.e. we could only apply this to the results in Fig 1 and not to the results

in Fig 2-3. Scaling the alpha has the advantage that we then can apply the same correction across all results, although not being the mathematically most correct choice.

We agree on showing R2 for the survey lake regressions. In the revision, we plan to put all regression details (R2s, coefficients etc.) in an appended table.

**Author's change: We have re-written the methods description to provide a clearer justification to the choices regarding how to deal with temporal autocorrelation. The text now reads (p. 9, l. 8-14): '*In order to not grant significance too generously, considering the temporal autocorrelation, we adjusted the α for relationships between WTT and response variables from 0.05 to 0.01. According to standard significance tables, it takes roughly twice as many observations to obtain significance of a correlation at the 0.01 level, compared to the 0.05 level, implying that an α adjustment from 0.05 to 0.01 approximately takes into account that only every second observation in the time-series could be assumed to be independent. Although it would be possible to explicitly include an autocorrelation term in the LMER models, the α adjustment was chosen since it could be applied in a systematic way to all regression results in the study, i.e. both to the LMER models and to the ordinary linear regression models.*'**

**Moreover, we now present full statistical details for the survey lake regressions in Table S1. We also plot all the raw data in Fig. S5, such that the readers can see that the relationships (when they occur) are linear.**

10. *3.1 Björntjärnarna chain lakes and Fig 1 Fig 1. The colours between Inlet stream and Övre Björntjärnen epi are almost indistinguishable. However, here I wonder about the inlet and outlet sites. First I wonder if it makes sense to add the inlet points to the analysis, since it is not affected by what happens in the lake. And about the outlet, I wonder to what WTT it is assigned to, since in the methods there is only a definition for the WTT of epi and hypo and inlet.*

Reply: We agree that the color fill of the inlet symbols need to be changed. This will be done in the revision.

The Reviewer is right that the inlet stream is not affected by what happens in the lakes. There might be a point with removing the inlet stream from the figure in question. For example, since we are using linear mixed effects regression models with site as random effect and WTT as fixed effect, the inlet site (which always has WTT set to 0) does not contribute to explaining any variance in the response variables (in terms of R2m). Thus, statistically, the inlet data does not play a role, in the sense that it neither contributes to nor removes explanatory value from the models. However, we think that displaying the inlet data serves a graphical purpose. It helps the reader get a better idea of the overall changes in DOM properties that happens in the catchment. Therefore we would like to keep the inlet data displayed. We do not see that there is a problem with keeping the inlet data as part of the statistics (although as mentioned it could also be removed without any impact on R2m).

As mentioned in response to point #3 above, in the revision we will clarify that complete mixing of the epilimnion is assumed, such that outlet water is equal in its properties (including WTT – time spent in lake) to epilimnetic water.

**Author's change: We have changed the color fill of the inlet data points to black.**

**However, as explained in the reply we did not chose to remove the inlet data. Removing it does not change the statistical output in any important way, and as we explain in the reply there are good reasons to show the inlet data as a starting point for the DOC that is then processed in the lakes.**

**Explanation to outlet WTT assumptions has been added on p. 6, l. 30 – p. 7, l. 2**

11. *P18 L 3-4: This may not be needed, since the y-labels are already shown in the plots, and the variables described in the methods.*

Reply: We will look into the author guidelines for Biogeosciences to see if it would be ok to remove these explanations to the variables or not. Perhaps the Reviewer is correct.

**Author's change: No change carried out. Although this can appear repetitive (defining variables), the figures with their captions should be informative as stand-alone units.**

12. *3.3 Survey lakes and Figs 2 and 3 In Fig. 2, the authors argue that there is a differentiated behaviour between brownwater lakes and clearer-water lakes. Even though this is later very neatly systematized in fig 3, I suggest adding this information somehow already in Fig 2, to help relate the plot with the description in the text. One suggestion would be to draw the lines in a colour indicating the corresponding DOC concentration, or a420, as reported in Table 1. This would also allow seeing if any two lines of epilimnion and hypolimnion are paired.*

Reply: We agree; this is a very good suggestion. We will change the graphics/color scheme of this figure to differentiate between clear and brown lakes (if possible we will display the full gradient between clear and brown lakes as suggested in the comment).

**Author's change: We have applied a color gradient from blue (clear lakes) to brown for the regression lines in Fig. 2.**

13. *On the other hand the two groups with opposite slopes, not only correspond to clear vs brown water lakes, but also they have very different ranges of variation of the WTT. For example, in Fig 2A, those lakes with negative slopes are also those with shortest ranges of variation of WTT. So here it is fair to wonder to what extent the slope is a statistical artefact resulting from the data not covering a similar range of variation. In fact, if all lakes were pooled together, the relationship between a254:a365 would be positive (we do not see the points in the graph, but I am joining the regression lines), and the same can be said for the other panels (if all points were to be pooled together, they would follow the trend described by those lakes with larger WTT ranges). With this I do not mean to invalidate the results, but maybe some more information could be added in order to emphasize the validity of these correlations, like adding the R2, or plotting their corresponding data points to evidence a clear linearity. I think it is important to solidify these results, as later they lead to intriguing interpretations like DOC concentrations increasing with longer WTT.*

Reply: As mentioned in reply to specific point #9 above, we plan to add expanded regression details (R2s, coefficients etc.) in an appended table, thereby presenting more complete statistical reporting in the new revised ms version.

In addition, to fully address the Reviewer's concern, we need to develop our discussion section with regard to what it means that our different lakes do not span the same range in WTT. The reviewer is correct that new (other) patterns would appear if data from the different lakes with different WTT spans would be pooled, but we argue here that such pooled patterns would be misleading. In fact, we see strong reasons not to pool data from the different sites as they represent ecosystems of fundamentally different character and functioning. First, the fast-turnover lakes have catchments that differ systematically in their properties compared to the catchment of the slow-turnover lakes, e.g. being much larger (0.79-3.2 km2 compared with 0.03-0.25 km2 for slow-turnover lakes) and having flatter areas with more wetlands in lower reaches close to the lakes, thus representing different hydrological functioning likely leading to DOM of different quality entering the lakes (Creed et al, 2015 Aquat Sc 72:1272-1285; Laudon et al, 2011 Ecosystems 14: 880-893). Secondly, the fast-turnover lakes themselves tend to represent a fundamentally different lake ecosystem type, i.e. brown-water, compared to the slow-turnover lakes (clear-water). Thus there is no doubt that our different study lakes represent lake ecosystems of different character, receiving water from catchments of different character. In other words, these systems are in many ways fundamentally different, and it is therefore not surprising that the dynamics of DOM composition indicators such as a254/a365 are different across these lakes.

In the revised discussion, we will expand on what it means that DOM quality variables show relationships that point in one direction in lakes that span a certain range in WTT, but point in another direction for lakes spanning another range in WTT. One possible explanation is that because lakes with different WTT ranges represent catchments that are systematically different, the DOM that enters the lakes in the different cases is of different quality and reactivity from start. For example,

short-turnover lakes receive water from large catchments with probable substantial wetland contributions. The colored wetland DOM may be relatively difficult to degrade (Berggren et al 2007 GBC 21:GB4002), so the lakes may stay brown even if WTT increases. Our slow-turnover lakes on the other hand receive DOM from smaller forest catchments with high hydrological connectivity. Such DOM may be more reactive in comparison (Laudon et al, 2011 Ecosystems 14: 880-893 – references therein) so these lakes easily get clear as WTT increases. Another possible explanation is that the response in DOM transformation processes to increasing WTT is in fact not linear. Initially an increase in WTT may lead to decreased a254/a365, but as WTT increases beyond a certain threshold the relationship reverses and a254/a365 starts to increase with WTT. Such non-linear dynamics would make sense in context of the ideas presented in Fig. 5, i.e. that clear-water and brow-water systems have different DOM transformation regimes, which opens up the possibility of passing thresholds that lead to regime shifts.

**Author's change: As previously mentioned, we now present full statistical details for the survey lake regressions in Table S1 and in Fig. S5. We have also added a new main section to the discussion (present section 4.4) that why the clearer lakes represent a different span in WTT compared with the relatively browner lakes. In short, the lakes that we included in order to obtain long water residence times are from a hydrological perspective functioning as forest kettle lakes, i.e. in the sense that they have no inlets, and they lack connectivity to riparian and wetland sources of DOC in their catchments. Such lakes are very seldom brown. The short-WTT lakes on the other hand have inflowing streams connecting them to riparian sources of DOC, and they are located in flat areas with direct connection to peatlands. Given the key role of riparian soils and peatlands as DOC sources to lakes, it is not surprising that these latter lakes are brown. The bottom line of our new discussion section is that (p. 14, l. 23-25) '*the lakes with long WTTs in this study may not be clear primarily because of their long water transit times per se, but because they are relatively disconnected from the key sources of colored DOC in the catchment, i.e. peatlands and riparian soils.*'**

14. *An interesting result that can also be drawn from figs 2-3, is the fact that those lakes with intermediate colour levels are less responsive to changes in WTT (slopes not significant). This would imply that this kind of lakes are less sensitive to hydrological variability and therefore less affected by hydrological events like rainfall or drought. This could also be mentioned/discussed in the text.*

Reply: We thank the Reviewer for an excellent suggestion, which we will add to the new discussion.

**Author's change: We now highlight both in the Results (p. 10, l. 21-23) and Discussion (p. 12, l. 3-5) that DOC properties in lakes of intermediate color were relatively less responsive to changes in WTT.**

15. *3.4 Experiments P 9 L 25: "similar to the changes observed over time" according to the caption of figure 3, it is not a variation over time but as a function of WTT.*

Reply: The Reviewer is correct, and we will change accordingly

**Author's change: The phrase 'over time' has been replaced with 'as functions of WTT' (p. 11, l. 6)**

16. *3.5 Overall color loss P9 L 29-30: "we multiplied the in situ rate of epilimnetic color loss in the survey lakes (same as slopes in Fig 2d) with the mean water transit time for the respective sites to find out how much total change there was in water color upon transit": I do not understand how this becomes a percentage of color loss, I suggest that this is more explicitly stated.*

Reply: Again, the reviewer is correct. In the revised version, we need to explain better how this calculation was performed. By multiplying the rate of color loss with WTT, we obtained total amounts of color losses during transit through the different lakes. We then normalized these amounts of color loss to the mean color of the different lakes.

**Author's change: Text has been modified, and reads: '*we multiplied the in situ rate of epilimnetic color loss in the survey lakes (slopes ± SE of epilimnetic $a_{420}$ regression models; see Table S1) with the mean WTT for the respective sites (Table*'**

***1) to find out how much total change there was in water color upon transit through each lake. The change was then expressed as a proportion (%) of the mean WTT for respective lakes.' (p. 11, l. 10-13)***

17. *Then, this "relative color change" is compared with the percentage change in the experiments. The calculation of the latter is never explained in the text, I suggest briefly explaining that.*

Reply: We agree, and we will change the manuscript as suggested. See response to specific comment #8 above.

**Author's change: See 'Author's change' in response to comment #8 above. This has already been handled, by addition of new text in the methods part, and by a small adjustment of the figure caption.**

18. *Discussion P10 L16-21: I think it should be taken into consideration here that the brownest lakes also had much shorter ranges of variation of WTT.*

Reply: As explained in response to specific comment #18 above, we will expand the discussion with regard to what it means that the brown-water lakes had lower WTTs than the clear-water lakes.

**Author's change: See 'Author's change' in response to comment #18 above. This has been dealt with by addition of the new discussion section 4.4.**

19. *P10 L 23-31 With the data set you have, including 3-7 year time series, you would not need to speculate about that. Why not just check how a420 and DOC change over time, or seasonally, in the inlet and outlet of the Björntjärnen lakes?*

Reply: Changed as suggested, and as explained in response to the major comment #1 above. By adding the raw data in an appendix plotted over "real" (calendar) time, we will be able to support the claims and speculations made here.

**Author's change: As previously mentioned, a new figure (present Fig. S4) has been added, plotting all raw data for DOC and its characteristics in the Björntjärnarna chain lakes over calendar time. We further highlight in the revised discussion that '*in our measurements of DOC concentrations and properties in the inlet stream to the Björntjärnarna catchment (Fig. S4), we found no systematic patterns with discharge*' (p. 12, l. 10-12)**

20. *Conclusions P13 L 2 "brown headwater lakes": or just "brown-water lakes"?*

Reply: Changed as suggested

**Author's change: Change implemented on p. 15, l. 18**

21. *P 13 L 4 "Thus change in WTT, e.g. due to a potentially wetter future climate, has no universal effect on lake color": This is a too hard statement, considering that your data for brown water lakes only covered a small range of WTT values.*

Reply: We will add a sentence after this statement clarifying that a possible limitation of the study was that the brown-water lakes only covered a relatively small range in WTTs. However, we do not believe that it is an over-statement that lake color is not universally affected by WTT.

**Author's change: We have expanded the sentence ending '*…has no universal effect on lake color*' with '*, at least not over the ranges in WTT that we studied*' (p. 15, l. 20-21)**

22. *Response to TECHNICAL COMMENTS*

*Some suggestions, even though I am not a native English speaker:*

*P2, L 26: ": : : as a result in temporal : : :": as a result of temporal*

*P3, L 1: "selective" instead of "selected".*

*P3, L 6: ": : : and analysed using linear mixed effects regression" probably not necessary to be mentioned at this stage. If it is mentioned, though, it should be stated what the mixed effects regression was for.*

*P5, L 8: "that passes by" not necessary.*

*P7, L 30: "The response of" instead of "The response in"*

*P7, L 3-4: "goes up" and "goes down": may I suggest avoiding these. Maybe that could be replaced simply for "If this ratio increases with WTT: : :" and "but if a420:DOC decreases: : :".*

Reply: Changes made as suggested

**Author's change: Suggested changes carried out on: p. 2, l. 28; p. 3, l. 7; p. 3, l. 12; p. 5, l. 21; p. 9, l. 2, and ; p. 7, l. 21-22**

> *In this work, the authors aim to determine the relevance of bio-and photo-degradation processes during the water transit time in individual lakes. The authors hypothesize that each process will prevail as a function of the color of the DOC compounds, so that biodegradation will target non-colored DOC while photo-degradation, colored DOC compounds. Using a complex data set at different temporal and spatial scales and including both field and experimental data, the authors found brown-water lakes to be dominated by biodegradation processes (not photo-degradation), which leads to their persistent brown-water color.*

> *The authors present these results as contrasting with the current paradigm of loss of colored constituents of DOC along the inland waters continuum. However, they do not provide such a continuum (i.e. accumulated water residence time along the landscape), they do not evaluate the molecular composition of DOC and, the presented here are net changes (i.e. including production and degradation of DOC) but they are not discussed as so. I consider the partitioning between photo and bio-degradation processes a key question to complete our knowledge on the pathways of C processing in inland waters. But because of this relevance, I ponder indispensable that the authors clarify those concerns above and the ones specified below (such as properly assessing the role of hydrology, improving the characterization of DOC or providing the complete results -the last specially affecting Figure 2-) before this manuscript can be considered for publication. I hope these comments are helpful and constructive.*

Reply: We are thankful for the Reviewer's constructive and much-thorough review that has helped us to improve the manuscript. We agree on the points mentioned in this general comment. Therefore, as explained in detail below (under specific comments), the revised paper will, compared with the original submission: 1) be more careful when discussing what our study suggests about loss of colored DOC along the inland water continuum; 2) discuss more explicitly what our results suggest about the molecular composition of DOC and the role of hydrology; 3) be clearer about the fact that our study addresses net changes in DOM properties; 3) provide appendices with more complete results, in terms of both statistical details and reporting/plotting of raw data.

All of the Reviewer's comments can and will be adequately addressed in our revised manuscript. However, the DOC characterization that we have at hand is limited to information that can be extracted from UV-VIS absorbance and compound-specific analyses performed using LC-MS. We get the impression that the Reviewer would have preferred to see additional DOC composition analyses (e.g. FT-ICR-MS molecular analyses or fluorescence EEM/PARAFAC), but such data do unfortunately not exist for this data set. Nonetheless, in the revision we will go deeper into the discussion of what our data suggest about patterns in molecular DOC composition. We will also provide justifications and explanations to why we present

and analyze the absorbance data the way that we do. We strongly believe that our manuscript has sufficient data to present an original and important story about how the properties of DOC change with transit time in lakes.

**Author's change: Changes are detailed below in response to the specific comments**

Response to SPECIFIC COMMENTS

1. *Abstract P1 L17: "photo-chemistry qualitatively dominated"…what does qualitatively mean here? That the changes in DOC quality were dominated by photo-decay? That you assess that in a qualitative (i.e. non-quantitative) way? Clarify in the text. Also, photo-chemistry dominated the DOC or the CDOM transformation in headwater lakes? How is the production of non-colored DOC evaluated? Clarify in the text.*

Reply: In the revised abstract, we have changed this phrase to clarify that 'changes in DOC quality were dominated by photo-decay', according to the first suggestion by the Reviewer. However, it is actually also true that we draw this conclusion based on a qualitative line of reasoning, i.e. we observed that the directions of change in the DOC quality *in situ* were matching the directions of DOC quality change observed in light exposure experiments (as opposed to dark conditions where the directions of DOC quality change were the opposite). In other words, we do not make a quantitative assessment here (e.g., % dominance by photo-processing), but rather we note the qualitative agreement between in situ and laboratory data. The revised methods description will be changed such that this becomes clearer.

**Author's change: With regard to the Reviewer's specific comments about the abstract phrasing, we believe that the concerns have been dealt with by rephrasing the sentence in question into the following: '*We found that influence from photo-decay dominated the changes in DOC quality in the epilimnia of relatively clear headwater lakes, resulting in systematic and selective net losses of colored DOC*' (p. 1, l. 7-18)**

**Changes in methods that are touched upon in our reply above are explained elsewhere (e.g., specific comment #6 by Reviewer #1). These do not specifically relate to what the Reviewer was asking for in this comment.**

2. *P1 L19: Was there a systematic relationship between color loss and WTT in Clearwater lakes? Add this information also.*

Reply: Yes, in clear lakes the color loss was systematic. We will add this information as suggested.

**Author's change: We now clarify that there were '*systematic and selective net losses of colored DOC*' (p. 1, l. 17)**

3. *Introduction P2 L17: Maybe biodegradation processes do not affect colored DOC preferentially, but that they do affect it at all has a stronger impact on the inland waters C budget than the consumption of in-situ produced DOC. Add information on the DOC sources and their relevance on the C budget here.*

Reply: This comment is not completely clear, language wise, but we think the Reviewer means that we should expand the text to explain that bacteria do indeed remove colored DOC – they just don't remove it as efficient as they remove non-colored DOC. In the revision, we will further mention additional DOC sources (we assume the Reviewer means 'other than terrestrial') and their relevance as suggested. We will cite one or a few references showing that boreal unproductive brown-water systems mainly have terrestrially-derived DOC, i.e. other sources play minor roles, although autochthonous production can be relatively more important in clearer and more nutrient-rich systems.

**Author's change: We have added the note that '*bacteria do consume colored humic substances at low rates (Tranvik, 1988)*' (p. 2, l. 16-17), but this does not change the point that we make in this sentence, i.e. that the biological degradation**

**of DOC appears as an unlikely a mechanism leading to selective color loss. Moreover, we mention photo-oxidation as a process that can lead to production of non-colored DOC (p. 2, l. 20-21). Finally, we acknowledge that '*non-colored DOC might be added by algae in productive waters*' using one of the existing references (p. 2, l. 12).**

> 4. *P2 L22: Available references on "efficient" photo processing, showing how polyphenolic, aromatic compounds are mostly affected by photo reactivity (assessed at a molecular level) in black and boreal waters, are missing (e.g. Stubbins et al. 2010 L&O, Kellerman et al. 2014 Nat. Comm. and references therein).*

Reply: We will insert the two suggested references. It appears most appropriate to cite these references after the statement 'UV light oxidation could theoretically explain losses of colored DOC', in the preceding sentence.

**Author's change: The citations have been inserted (p. 2, l. 21)**

> 5. *P2 L27: I agree with the authors that the assessment of the variability of WTT within systems is very relevant. However, without assessing how that variability is linked to changes in color of runoff DOC, it is hard to attribute the changes in the lake just to insitu biogeochemical processing. Clarify that here and incorporate that perspective throughout the text -see comments below-.*

Reply: We will re-write this section to clarify that the export of DOC from small headwater catchments in the region is strongly episodic. There are several classical papers from the Krycklan Catchment Study to exemplify this; for example Laudon et al (2004 Aquat. Sci. 66:223-230) showed that 50-70% of the entire annual organic carbon export comes just during a short period of snowmelt in spring, and we know that much of the remaining export happens during discrete autumn rains. Given this pulsed nature of inflowing water and DOC, we do not agree with the Reviewer that it matters how the DOC or color varies temporally during other situations than high-flow. If the total DOC export is negligible during low-flow, then this carbon will not contribute significantly to the DOC that resides and gets processed in the recipient lakes, and thus it is not relevant to know the properties of such DOC entering during low-flow. It would be more critical if there is large variability in DOC concentrations and color within the high-flow episodes, but this does not appear to be the case.

We do agree with the Reviewer that we need to incorporate this perspective better, both here in the introduction and elsewhere in the manuscript. In the new revised introduction and methods parts we will explain why we expect that (in our specific study lakes) it is the transit times though the lakes that will matter for the color – not differences in color levels of the water that comes in from the catchment during different times. Moreover, we will test and confirm that this assumption is true, as explained in response to specific comment #25 below, with added results/discussion parts related to this. For example, we will present data showing how much (%) of the total DOC exports that takes place during episodes, defined as flow rates above a certain percentile. We will be able to show that: 1) most of the DOC and color enters the lakes during high-flow conditions and; 2) DOC and color variations are relatively small during these high-flow conditions. Together these two circumstances imply that colored DOC enters the lakes mainly in distinct high-flow pulses, and it is removed during in-lake processing during low-flow periods when the catchment plays a negligible role in adding new DOC and color to the lakes. See more details in our response to specific comment #25.

**Author's change: We have added two sentences to the revised introduction, the first one clarifying that '*the DOC needs to enter a lake in distinct pulses, each time with similar concentrations and chemical properties, such that subsequent DOC quality changes in the lake are dependent on the WTT*' (p. 2, l. 31-33). In the second new sentence, it is explained that '*We chose lakes … around the Krycklan catchment in northern Sweden, where there is extensive research in support of the assumption that DOC export is highly episodic, with pulses that generally bring DOC of similar quantity and quality during peak flow (Laudon et al., 2011)*' (p. 3, l. 3-5).**

**Changes that relate to the second paragraph of our reply are explained under specific comment #25 below**

6. *Methods P3 L16: modify this sentence into "lakes are located in the boreal region, where nutrients" and provide a reference of that distribution.*

Reply: We will make the change as suggested and cite the distribution by Verpoorter et al (2014, GRL 41: 6396-6402).

**Author's change: Change implemented on (p. 3, l. 22-23).**

7. *P3 L30-32: Although, low effects of pH on the optical properties of DOM have been reported at the most frequent inland water's pH range (i.e. 5.5-7.5), they can be important at lower pH values (< 4.5), such as the ones included in this study. Accordingly, add a paragraph in the discussion stating which lakes presented these low pH values (i.e. 3.4) and how could that affect your absorbance measurements (some useful literature: Pullin and Cabanis et al., 2003, Geochim. Cosmochim Act.; Patel- Sorrentino et al., 2002 Wat. Res.; , Spencer et al., 2007, Wat. Res.).*

Reply: We will add this discussion as requested. The Reviewer is correct that there can be optical effects due to low pH values, and that we presently do not give attention to such effects. In principle, as explained in these references that the reviewer provides, an extremely low pH causes a very high degree of protonation of the molecules, which in turns means that they physically shrink into a compact mode. In their most compact/protonated state, the overall light absorption by the DOC molecules may not be at the highest, but specifically the short-wavelength UV radiation that has most energy is efficiently absorbed. This can lead to marginally higher photo-reactivity at an extremely low pH compared to moderately acidic conditions. We will discuss how this might have influenced our results in the new manuscript version.

**Author's change: We have re-written the text in question to acknowledge that absorbance at certain wavelengths '*can also be enhanced by an extremely low pH (~4), potentially leading to higher photo-reactivity (Anesio and Granéli, 2003;Gennings et al., 2001), although these effects appear relatively small compared to those at high pH (Pace et al., 2012)*' (p. 4, l. 6-8).**

8. *P4 L6: Consider reporting Catchment area/ Lake area ratio as a more relevant variable to discuss epilimnetic WTT than catchment area alone.*

Reply: Since our study sites are similar in size, it is mainly the catchment area that is important for the WTT. To be more precise, the variation in lake area (1-5 ha) is small compared to the 100-fold variation in catchment area (Table 1). Therefore, we do not consider that it is necessary to also report catchment to lake area ratios. We will explain this in the revised ms version.

**Author's change: Following text has been changed: '*While the lakes have small areas, varying only between 0.01-0.05 km$^2$, the catchment areas vary 100-fold from 0.03 km$^2$ to 3.25 km$^2$, resulting in...*' (p. 4, l. 15-16). In this way we emphasize that it is the catchment area that is important for WTT, rather than lake area or the ratio between catchment area and lake area.**

9. *P5 L3: Why using only 3 wavelengths if the whole spectra were available? Given the aim of the study, much more robust conclusions could be reached if other widespread descriptors such as SUVA254 and slope analysis were included, and I recommend their inclusion. Those descriptors are widespread, and in particular, spectral slope analysis, is recognized to provide further insight into DOM composition than absorption coefficients alone (see Helms et al. 2009 L&O, Loiselle et al. 2009 L&O). Package "cdom" in R could be a useful tool to perform that exploration.*

Reply: We agree that SUVA254 is a relevant variable, and we did use this in previous manuscript versions. However, since SUVA essentially showed the same patterns as the a420/DOC ratio, we removed it to avoid redundant data that does not add to the story. We will explain in the revised version that these two variables are strongly correlated. Similarly, while we could use a number of different spectral slope indicators, it would not be meaningful since all of them would correlate strongly with the spectral slope indicator that we already have, i.e. the a254/a365. However, what we can do in the revision is to explain

better why certain choices were made, and what these choices mean. Part of this choice is a matter of research tradition, or even taste, but we think it is important to address how the metrics that we have chosen relate to other metrics that are common in the literature. Thus, we will add such explanations to the revised ms version.

**Author's change: Regarding our choice of the a420/DOC ratio, we now explain that '*also other indices of color per unit DOC are common in the literature, especially specific UV light absorbance at 254 nm (SUVA$_{254}$). However, in our study the overall relationship between SUVA$_{254}$ and a$_{420}$ : DOC was strong and linear (r$^2$ = 0.80, n = 680; all sites and sampling dates), and the two variables showed the same patterns in response to changing WTT. Therefore, to avoid presenting the same patterns twice, we do not report SUVA$_{254}$ in this paper*' (p. 7, l. 25-27). We also explain that '*There are many … spectral slope indices in the literature essentially providing the same information, but we chose a$_{254}$ : a$_{365}$ since it is a simple index that has been used previously in the study area (Berggren et al., 2007a;Ågren et al., 2008a).*' (p. 7, l. 14-15)**

    10. *P5: Calculations for outflow are nor provided but they are presented in Figure 1. Add this information here.*

Reply: In the revision we will clarify that complete mixing of the epilimnion is assumed, such that outlet water is equal in its properties (including WTT – time spent in lake) to epilimnetic water.

**Author's change: Author's change: Explanation to outlet WTT assumptions added on p. 6, l. 30 – p. 7, l. 2**

    11. *P6 L17: Are all the other catchments spatially independent? Even if the inlet streams are considered negligible, what about the accumulated time in the catchment (sensu Müller et al. 2013 Aq. Sci.)?*

Reply: With regard to the first question: yes, all other catchments are spatially independent. Regarding the second question: we are interested in the accumulated time in the freshwater network itself, sensu Berggren et al (2009, L&O 54:1333-1342). This is in our case the same as the accumulated time in the view of Müller et al. (2013), because the streams are headwaters even in the strictest definition, i.e. there are no upstream lakes that would add residence time. Thus the drainage dynamics is strongly pulsed, and water is flushed more or less directly from soils to the lakes. These aspects will be explained in our revision.

**Author's change: The following clarification has been added: '*there were no upstream lakes in the catchments, in addition to the study lakes themselves, implying that the drainage represented true headwater sources directly from surrounding soils*' (p. 6, l. 28-30)**

    12. *P6 L27: The relative contribution of LMWC to total DOC (%) should be used instead of the total concentration of organic acids. A higher total sum of organic acids could be just due to a higher DOC concentration. Thus, to clarify if samples have a higher relative contribution of LMWC compounds or just higher DOC, the relative contribution of LMWC to total DOC (%) should be used, and ideally both (LMWC for each sample and in % and in mgC L-1) shown in the Supplementary Information. Also, is the correlation between a254:a365 and the organic acids positive or negative? Should be stated.*

Reply: In the revised ms, we will show LMWC both as absolute amounts and as percentages of total DOC. These variables will show similar patterns. In the new manuscript, we will clarify that the correlation in question is positive, as suggested by the reviewer.

**Author's change: The correlation between a254:a365 and relative LMWC (% of DOC) concentration has now been added to the present Fig. S3. We have also edited the manuscript text on p. 7, l. 17-20 in the following way: '*In agreement with the expectations based on Berggren et al. (2010), a$_{254}$ : a$_{365}$ in this study was positively correlated to both absolute (mg C L$^{-1}$) and relative (% of DOC) total concentrations of low molecular weight carbon compounds in the form of organic acids, free amino acids and simple carbohydrates (Fig. S3)*'**

13. *P7 L13: Bacteria might dominate the biomass, but still be predated by heterotrophic flagellates. How does the bacterial abundance looked during the experiments? Moreover, 450 days is a very long period, which effects would have both the predation and the death of the bacterial community and subsequent mineralization of that biomass on the results? How fair is it to consider that these results reproduce the biodegradation process occurring in the field where lakes behave like chemostats not like batch incubations? Justify in the text, and discuss later the implications and assumptions that have to be done to compare both results in the discussion.*

Reply: Since there is an overlap between this comment and concerns by Reviewer #1, we like to start by pasting part of the reply to specific comment #6 by Rev 1:

"This comment helped us see that the purpose of our laboratory experiments was not sufficiently well described in the original submission. Briefly, what we wanted to achieve was experimental conditions during which either 1) photochemical reactions strongly and dominantly influenced the DOM transformation, or 2) microbial degradation strongly dominated the DOM transformation. Thus the experiments were designed such that the response to a large light dose or a long microbial process time in the dark was measured. While we don't believe that such experiments mimic lake *in situ* conditions in an adequate way, they do provide qualitative information about how the DOM responds to the isolated effects of photochemical and biological decay. Interestingly, the patterns of DOM transformation found in dark experiments well matched the *in situ* DOM quality changes observed in dark (brown or hypolimnetic) environments, while our light experiments matched the qualitative patterns in DOM transformation in clearer and more light-exposed environments *in situ*. These findings are supporting our interpretations. […] Therefore, based on the above, in the revised manuscript we will provide a clearer rationale for the experimental design of our study. We will also highlight that the experimental results only provide qualitative information about how the DOM responds to different types of decay – it is not possible to make quantitative comparisons."

On the specific comment about biomass, the present discussion paper cites Daniel et al (2005) on the rough biomass contribution of 90% by bacteria in food webs (microbial communities) developed in the dark in humic water. It is a reasonable assumption that bacteria were similarly abundant in our incubations, but as we did not measure biomass this can only be speculated on. We did monitor bacterial production (not shown), and as expected it decreased systematically with increasing incubation time. This is much expected as bacterial production has been shown to decrease with increasing water residence times *in situ*, in lakes of the study area (Bergström & Jansson 2000, Microb Ecol 39:101-115; Berggren et al 2009, L&O 54:1333-1342).

In the revised discussion we will give attention to the fact that our incubations involved batch DOM degradation performed by an artificial microbial 'bottle community' that may be different from the in situ community. We will however maintain that these dark incubations fulfilled their purpose of showing how (qualitatively) DOM properties change in response long-term biological processing.

**Author's change: With regard to the general experimental design, the following text has been inserted to justify and explain our approach better: '*We performed laboratory experiments on water from three catchments to disentangle the isolated effect on DOC quality by UV light degradation from that of microbial processing. The purpose with the experimental design was to create conditions during which either 1) photochemical reactions strongly and dominantly influenced the DOC transformation, or 2) microbial degradation strongly dominated the DOC transformation. Therefore, we measured the DOC quality responses to a large light dose or a long microbial process time in the dark. However, the experiments were not designed to reflect in situ decay rates.*' (p. 7, l. 29 – p. 8, l. 1)**

**Further we have added mentioning of that bacterial communities likely were different in the bottle experiments compared to in the field (p. 8, l. 2-3)**

**With regard to the Reviewer's comment about biomass, we have clarified that the biomass assumption is based on a reference and not our own measurements. The text now reads '*it can based on Daniel et al. (2005) be assumed…*' (p. 8, l. 9)**

14. *P7 L18: I agree that microbial processing can happen in the entire water column, but I believe the simultaneous action of UV and biodegradation cannot be discarded. On the one side and mainly, because photo-mineralization rates are faster than biodegradation rates. On the other side, because there are several situations where the entering water will be exposed to both ( i) water in the hypolimnion, would have been initially exposed to both UV and microbes when entering the lake, ii) under ice conditions, microbial activity would also be minimal due to low water temperature iii) during the ice-free period and at that latitude, daylight is almost for 24h). Thus, both processes are likely to occur also simultaneously or following the inverse sequence (photodeg --> biodeg). Justify that, considering the number of papers using the opposite approach. The authors could also perform a much deeper exploration of the changes between layers with the temporal data available and in light of the results shown in Fig.2 on that direction.*

Reply: This is a relevant point brought up by the Reviewer – there are certainly numerous interactions between microbial and photochemical processes in nature, but with our experimental approach we are not able to address these interactions. As mentioned in response to the preceding comment (#13), we plan to expand the discussion with a section that deals with limitations in the experimental approach that we chose for this study. In this new section we will also bring up the aspects mentioned in the comment above, i.e. potential interactions between light and dark processes that we currently do not recognize in the discussion paper. We will link this discussion to what results from the different layers, as hypolimnetic waters have very little light intrusion also in the clearest of the sites. Thus differences in patterns between the depth strata of the same lakes can be used to discuss the impact of the light processing *in situ*.

**Author's change: We have re-phrased the commented text and added a reference to clarify that '*although photo- and bio-degradation can happen simultaneous, much of the DOC likely has stayed variable and potentially long time in darkness before getting in contact with UV light (Gonsior et al., 2013)*' (p. 8, l. 17-19). Moreover, as mentioned in Author's change in response to comment #13 above, the experimental design is now explained clearer, i.e. that it is the individual impact of microbiological and photochemical processes, respectively, that we aimed at resolving – not the interactions.**

**Discussion of potential interactions between these two types of processes has been added on p. 13, l. 26-32. Here we discuss that these two kinds of processes often interact positively on DOC decay, which might suggest that lakes of intermediate color have the most efficient DOC loss because in those lakes the two processes appear to be in balance.**

**We finally wrote in the reply above that we would expand the discussion regarding results from the different water layers. However, after a closer look, we see it was emphasized already in our discussion paper that the degradation processes showed different patterns in epilimnetic and hypolimnetic waters, especially color loss and photo-decay being coupled specifically to epilimnia (see e.g., p. 13, l. 16). Therefore, no further changes were made.**

15. *P7 L25: Similarly for photo-decay than for bio-decay: even if a radiation equivalent to two years was applied, there was no water renewal considered. Discuss how well you expect this results to reproduce the process in the field.*

Reply: This will be discussed as suggested. Again, the Reviewer is right that we did not perfectly reproduce *in situ* conditions during our experiments. However, the pulsed nature of DOM input to the lakes makes the *in situ* processing function in a similar way as 'batch processing'. Thus a similar response could be expected.

**Author's change: As explained in response to point #13 above, a new text has been added (p. 7, l. 29 – p. 8, l. 1) that explains better our experimental design and approach. Since the intention was not to perfectly reproduce *in situ* conditions, we believe that it is not necessary to dive deeper into this discussion.**

16. *P8L8: Where the assumptions fulfilled?*

Reply: Yes, it was fulfilled since there was generally no temporal autocorrelation for two time steps. We will re-write this section to clarify our approach as explained in response to specific comment #9 by Rev #1.

**Author's change: See Author's change response to specific comment #9 by Rev #1**

    *17. P8L11: Specify which variables are set as the fixed effects and as the random effects here.*

Reply: We will specify that WTT is the fixed effect and site is the random effect.

**Author's change: Change implemented on (p. 9, l. 3-5)**

    *18. Results P8 L24: Is "the most dynamic lake" also the smaller lake (volume)? The one with bigger catchment? I missed that in the discussion later and to discuss the controls on the trends on WTT and color in the epi- and hypolimnion.*

Reply: We will remove this mentioning of 'most dynamic' and 'least dynamic' lakes as it could be misinterpreted. Moreover, we will clarify that lakes with large dynamics spans in WTT are generally those that have intermediate turnover times. These lakes can build up long residence times during extended dry periods, but when an exceptionally large discharge pulse comes, then much of the water can be renewed and the WTT may drop dramatically. In our case it is not so much the lake size that determines the WTT (all lakes are small) but rather the catchment size.

**Author's change: We removed the parts about 'most dynamic' and 'least dynamic' (p. 10, l. 2). The discussion parts that the reviewer missed is included in the new lines on (p. 14, l. 11-25)**

    *19. P9 paragraph 3.4: There are no details provided on what is considered "change" in the incubations. Also, changes in DOC and ideally DOC decay rate should be shown in Fig. 3*

Reply: With regard to the comment about lacking explanation to how 'change' was calculated, the Reviewer is correct, and we will change accordingly. When it comes to the DOC decay, we need to stress that these incubations were not performed for quantification purposes, but only for seeing the changes in DOC quality upon light irradiation and biological decay respectively. Thus we do not consider that it is relevant to add decay rates to Fig. 3, which would remove the focus from what is important in this Figure, diluting the message. However, we will include more details about the incubation decay elsewhere in the manuscript, in the results text (at least ranges) and possibly in the supplemental materials.

**Author's change: The following text was added to p. 8, l. 30-33: '*The DOC and absorption coefficients were analyzed before and after each experiment, and the changes from the beginning to the end of the incubation (final value minus start value) were calculated for the absorbance ratio $a_{254} : a_{365}$ and the carbon color indicator for $a_{420} : DOC$. In addition, the relative change (%) in color ($a_{420}$) from beginning to end of the incubations was calculated and used for a qualitative comparison with percentage loss in $a_{420}$ during water transit through the different lakes in situ*'. Additionally the Figure 3-4 captions now mentions how the change in rates during experiments was calculated from beginning (start) to final (stop) values.**

    *20. P9 L30: Provide details (e.g. units) of this calculation. Also, only the ones in Fig. 2 were included, or all the sites? Clarify. Also, looking at these figures, how does the reader know which are the "clearest" and "darkest" lakes? different symbols should be used. Moreover, that categorization should be clearly defined and the cut-off between both justified previously and based on values previously reported in the literature. Also, in Table 1, it should be an additional categorical variable stating if a lake is "clear" or "brown".*

Reply: All these changes will be carried out as suggested. All of the sites were included, as will be explained.

**Author's change: We followed the suggestion in specific comment #12 by Reviewer 1. Therefore, instead of categorizing the lakes as clear or brown, we have applied a color gradient from blue (clear lakes) to brown for the regression lines in Fig. 2. Because we now consider the full gradient, there is no point in adding a category column to Table 1.**

21. *Discussion P10 L10: Which impact could it have that WTT does not span a whole hydrological year? Discuss here.*

Reply: This means that much of the entire lake volume is renewed during the snow melt period alone (since typically a majority of the entire annual water budget is flushed out at this time). During the low-flow period that follows in summer the lakes will typically act as a reactor that carry out batch processing of 'spring flood' water. We will discuss this in the revised paper.

**Author's change: We have clarified that this means that 'In these lakes the major discharge pulses alone (snow melt and storms) renew all the water at least once per year, and the processes that removes colored DOC are too slow to result in significant color loss in between of these discharge pulses' (p. 11, l. 23-25)**

22. *P10 L13: "the quantitative photo-bleaching in the Björntjärna catchment", what do the authors mean? Was there a quantitative evaluation of that? What is the total DOC photo-bleached in the catchment? Also were those studies (Lindell et al. 2000; Vachon et al. 2016) using a similar approach?*

Reply: We will remove the word 'quantitative' as it causes confusion. We will also change the word 'catchment' to 'lakes' as this is a typographical error. With regard to the cited references, we do not claim that these used a 'similar approach' in relation to our study or in relation to each other. We merely point out that that these studies suggest that 'recent inputs of humic materials from the catchment represent a relatively photo-reactive DOC source'.

**Author's change: Changes carried out according to the above reply on p. 11, l. 21 and 26**

23. *P10 L17: If I am correct, now comes the only available definition of "brown" lakes. Also...what other variables define a brown or clear- water lake?? Could the authors relate these categories with e.g. morphological variables? (e.g. volume, catchment/lake area, peatland presence, etc). It feels somehow poor to discuss the change in color using a categorical variable built upon that same parameter. I recommend to provide a full multi-parametrical characterization of the two groups.*

Reply: We agree with the Reviewer. In the revision, we will bring in more catchment descriptors into Table 1 (peatland presence, morphometric indices) and present/discuss the gradient from brown to clear lakes in a multi-dimensional way. In short the browner lakes are those with larger catchments and thus larger catchment areas to lake areas. However, also peatland cover might contribute to color, which we will discuss more clearly as suggested.

**Author's change: Because we followed the suggestion in specific comment #12 by Reviewer 1, the categorization of lakes into 'clear' and ''brown has now been removed or strongly toned down in the manuscript. Therefore, this comment becomes partly redundant. Nonetheless, more details on how the browner short-WTT lakes differ from clearer long-WTT lakes in terms of catchment properties are presented on p. 4, l. 20-22 and discussed on p. 14, l. 16-25.**

24. *P10 L20: Müller et al. 2013 evaluated the influence of lateral water inputs. Could later inputs explain the patterns found here? Was there some assessment of lateral fluxes in the systems (e.g. groundwater inputs) so as to discard that from happening in some of the other brown-water lakes?? Discuss in the text.*

Reply: In our analysis no distinction is made between diffuse and inlet stream fluxes. It is assumed that the entire catchment contributes with the same areal runoff to the lake, as explained in the supplementary methods. Four of the lakes have no permanent inlets, so here the groundwater inflow is up to 100%, but in the Björntjärnarna lakes there are inlet streams draining ca 90% of catchment. All cases, however, fall under the same assumptions.

We agree with the Reviewer that the discussion should bring up the possible impact of groundwater inflow more clearly. Possibly in a site like Stortjärnen (the lake in which color and DOC increased during low flow, where there is no permanent inlet but instead large amounts of peat with diffuse flow paths around the lake), we might be underestimating the amount of water and DOC that enters during baseflow. This aspect will be added to the revised discussion.

**Author's change: Groundwater input via surrounding peatlands is now mentioned as possible explanation to why DOC and color increased with WTT in Stortjärnen (p. 12, l. 25)**

25. *P10 L30: How is it in Fig. S1b evaluated the contribution of runoff to total water and DOC? The authors do not explicitly evaluate this and they should do so. According to that figure, as runoff increased, WTT decreased. Therefore, we could expect the exported water/DOC during episodic flows to be flushed away from lakes also. As WTT turns longer after the flow, the DOC sources and thus composition, should also recover. To avoid that interpretation, the authors should explicitly evaluate the contribution of runoff to the budget, and discuss more in depth differences found in that sense between the different type of lakes (i.e. above and below one hydrological year, clear and brown) and their layers (epi vs. hypolimnion).*

Reply: We thank the Reviewer for pointing out this weakness in our manuscript. Indeed it is not clear from Fig. S1 how important hydrological episodes are for DOC input to the lakes. Because the figure is integrating a lot of data, the pattern appears smoothed out, and the readers cannot clearly see how the episodes play, especially not in fall.

We will follow the suggestion and report numbers saying how much of the total DOC budget that entered the different sites during different types of hydrological situations (at different flow percentile ranges, parts of the year etc.). We will also discuss whether or not high-flow water was flushed away from the lakes, as mentioned by the Reviewer. In short, the assumption that we make is that outflow is equal to total inflow, implying that some of the water that enters the mixed layer always will be flushed out. However, the major annually reoccurring high-flow events happen during parts of the year when the lakes are non-stratified (spring and autumn) which means that this inflowing water will mix with the entire lake volume and thus is relatively less likely to be flushed out compared to inflowing water in summer moving through the epilimnion.

**Author's change: In the revision, we report in the results that '*For all catchments, 50% or more of the annual runoff was represented by discharge above the 90th percentile, i.e. a majority of the discharge happened during hydrological episodes*' (p. 9, l. 27-28). This is brought up again in the discussion on (p. 12, l. 15-16), and a reference to the new Fig. S4 is here made to support the point about low flow periods being relatively unimportant. The Fig. S4 shows raw data on flow with DOC and other things plotted in the same panel.**

26. *P11 L13: I consider the authors cannot conclude this, as there cannot be confident on the evaluation of the inputs performed, and that should be discussed at that point. Thus, "DOC accumulation can overcome degradation even in some small individual unproductive lakes" and it can be due to reduced degradation or to lateral terrestrial inputs. Add that discussion.*

Reply: The Reviewer is correct. We will add the suggested phrase and the potential different explanations that the Reviewer brings up.

**Author's change: The suggested phrase has been added in the sentence on (p. 12, l. 30), and the sentences before this one brings up the different possible causes. For example, groundwater input via surrounding peatlands is now mentioned as possible explanation to why DOC and color increased with WTT specifically in Stortjärnen (p. 12, l. 25)**

27. *P11 L17: The authors should evaluate these processes always as a net result of production vs consumption. Thus, in brown-water lakes, the apparent decrease in LMWC is due to consumption above production. Opposite would hold true for Clearwater lakes. Implications of acknowledging that are apparent and results need to be discussed under that light.*

Reply: The fact that these processes are a result of net production vs consumption will be mentioned here as suggested.

**Author's change: Changes carried out as suggested on p. 13, l. 7 ('*fractions were consumed more (and/or produced less)*')**

28. *P12 L1: Thus, the total color loss might be the same in both type of lakes, but the relative loss in brown water much lower. So... if the brown water lakes correspond to the headwater and lower WTT lakes, terrestrial inputs being more important and frequent (lower WTT), could that color loss in brown lakes (even if just representing a small fraction of the total color) be indeed more important at the landscape level? Discuss, and as previously stated, provide a better characterization (including morphology and relation with the catchment, especially with terrestrial inputs) of the two lake types (clear vs brown).*

Reply: Based on our actual data, it is difficult to push the discussion into the direction that the Reviewer suggests here. However, we can change the discussion to highlight that it is possible that color loss in brown-water lakes is more important at the landscape level than what it appears to be in our study lakes.

**Author's change: As hinted in the reply, this discussion was not easy to fit in. However, we believe that the new discussion section 4.4 covers what the Reviewer is asking for. Here it is explained how the different types of lakes differ in terms of catchment properties, hydrological inputs and possible DOC sources. (e.g., p. 14, l. 16-25)**

29. *P12 L20: What does it mean that it eventually "takes over"? Which mechanism could then explain it? Are there no other environmental or morphological factors that can explain that? Which could be the temporal threshold and could that be related with the hydrology? Include these questions in the discussion.*

Reply: We agree that the phrase 'takes over' is unclear, and it should be removed. What we mean is that the threshold is passed when the directions of DOM quality change reverse as shown in Fig 3a-b. Somewhere around the a420 of 7 m-1 there is a change from DOM processing characteristic of dark conditions (biology) to DOM processing characteristic of light conditions (photo-chemistry). It is very clear in Fig 3a-b that the 0 line is crossed at a certain distinct point. A long extended period of low flow could possibly induce passage of this threshold. It would then be expected that color is lost at an accelerated speed. However, a new high-flow episode with brown water entering the lake could push the system back across the same threshold again, into the brown-water state. We will develop this discussion in the revision.

**Author's change: We decided to remove the whole sentence instead on expanding on this. With the new discussion section 4.4, adding the suggested discussion points here would either make the text repetitive or obvious/redundant.**

30. *P11 L23: I believe it is very bold to interpret the incubation results that way. They give us an idea of the changes caused by one mechanism, but they do not exclude other mechanisms to happen. All the potential processes that could produce these changes in in-situ lake CDOM should be discussed.*

Reply: We agree with the Reviewer again. However, we did not intend to claim that "excretion of humic-like chromophoric molecules by bacteria" is the only process that can produce CDOM in lakes. Moreover, we do not propose that this specific process is significant, because this we do not now. The idea was just to put all cards on the table and mention this as a possible mechanism that might have played together with several other mechanisms. We will tone this part down further, to not give the readers the idea that we suggest bacterial color excretion to be major. Instead we will link this discussion more clearly to other possible causes of CDOM increase.

**Author's change: We now refer to the microbial excretion of humic-like chromophoric molecules as 'hypothetical' and 'less likely' (p. 13, l. 8-1)**

31. *Summary and conclusions The first sentence sounds contradictory. If only headwater lakes are being evaluated, then, it cannot be assessed a general freshwaters pattern. I believe the fact that headwater streams present "a sustained*

*level of pigmentation regardless of WTT variations" is extremely interesting, and the relationship of that with hydrology and input sources deserves a much deeper exploration, and I encourage the authors to move towards that direction. Otherwise, the affirmation that "the results may not conform to the general reported pattern of selective removal of colored constituents" without providing an evaluation of the DOC sources variability, does not hold firmly.*

Reply: We will change the phrasing to make it even clearer that we do not propose a general freshwater pattern based on our study. We consider that it is relevant to contrast our findings with other studies showing continuos color loss along the freshwater continuum. However, our point is neither to refute such previous studies, nor to suggest new dynamics for the whole land-sea continuum. Our results have important implications for the color dynamics of small headwater lakes, but this is where the scope of our study ends.

**Author's change: We have changed the phrasing to '*our results exemplifies how individual brown-water lakes may not conform…*' (p. 15, l. 18)**

    *32. Tables and figures Table 1: Provide volume or depth information. Provide the categorical variable: clear or brown.*

Reply: Changed as suggested

**Author's change: Mean lake depth added to Table 1. The category variable is however redundant now when we have removed the categorization from the paper.**

    *33. Figure 1: use different symbol for inlet or black color, it cannot be distinguished. Also, add definition of the outlet calculation in methods. Without that information… Shouldn't "out" WTT be longer than "epi" WTT? Answer and clarify in the text.*

Reply: Changed as suggested. See also response to specific comment #10 above regarding the outlet WTT.

**Author's change: Inlet symbols in Fig. 1 have been changed to black as suggested. See response to specific comment #10 above regarding the outlet WTT.**

    *34. Figure 2: I recommend fully re-working this figure and splitting it in two if needed. Above all, all data should be provided, for all lakes and layers, significant or not, so that the relationships not shown here could be evaluated by the reader. Moreover:*
*- The reader should be able to identify the lakes, to assess if the trends in the two layers are opposed or similar in each system.*
*- Also, it is impossible to assess the adequacy of the fittings without the points even if p-value is reported, and that is very important information.*
*- It is not clear which are the clear and which the brown water lakes, include that information in the legend.*
*- There seems to be two groups also as a function of WTT, how does that influence the results? e.g. in Fig 2d, where epilimnion and hypolimnion present completely opposite trends for the two age groups.*
*- Consider providing a summary table with the results of all the regressions, so the reader realizes how many fittings and which were not significant also.*

Reply: In the revised files, we will provide a table with detailed regression details (coefficients R2 values etc) for all the different relationships. We will also denote clear and brown lakes (or if possible the whole spectrum) in Fig 2. If the lakes would not be individually identifiable in the figure itself, then at least they will be so in the supplementary material.

However, adding all raw data to Fig 2 points will not be possible as the figure will become a complete mess with so many scattered points. Instead, we can show the individual relationships with raw data points in the supplementary information.

The fact that epilimnetic and hypolimnetic patterns sometimes are opposite is something that is already brought up in the results, e.g. section 3.3. However, we agree with the reviewer that this could be given more attention, especially in the discussion. For example, hypolimnia are darker, so it is not surprising that changes in DOM properties down there may be indicative of dark microbial DOM processing even in clear lakes.

**Author's change: Everything of what the author asks for here is included in the new Fig. S5, new Table S1, new discussion section 4.4, and the change in Fig. 2 with different color fill of the symbols.**

35. *Figure 4. It is not clear how that % is calculated (see previous comment). Also, are these changes significantly different from zero? Add that information as well as a zeroline. Clarify also in the caption that the slopes correspond to the ones in Fig. 2d. The reader should be able to identify to which line in Fig. 2d corresponds each dot in Fig. 4, modify accordingly.*

Reply: We thank the Reviewer for pointing this out. Explanations and the zero line added as suggested.

**Author's change: Zero-line added and the caption has been re-written to make all suggested corrections. However, for significances and for identification of the individual lakes, the reader is referred to Table S1 and Fig. S5 – not to Fig. 2.**

36. *Figure 5: The presence and contents of this figure should be re-evaluated once the suggested changes have been taken into account. Also, as it reads now, it is a bit like the chicken or the egg dilemma: are brown regime lakes brown because they have high water color? Or do they have color because of their brown regime? In other words, what is the progress on defining color regime only based on color?*

Reply: We believe that we already have an extensive discussion related to the 'chicken/egg' dilemma in section 4.4 of the discussion paper. However, we could highlight even clearer the key importance of the color of the inlet water for the trajectory of any given lake. Another aspect that plays is the degree to which the lake water is renewed during the spring flood. For example, if a lake annually is filled with spring flood water black as coffee, there is no room for dynamics that would allow such a lake to develop into a clear-water lake. Conversely, if only a small part of lake water is renewed annually, and if the inlet water itself is relatively clear, then it could be expected that the lake would remain clear at all times. In cases between these two extremes, we would expect to see more dynamics and shifts in color and DOM processing. In the revised section 4.4 we will discuss this deeper.

**Author's change: With the new discussion section 4.4 and other changes carried out to improve the discussion in response to various above comments, we believe that no further change is required.**

Response to TECHNICAL COMMENTS

*P1 L13: "DOC quality and color"…if color and quality are considered separately, which variables are being used to describe quality besides absorbance? Isn't color quality of DOC? I suggest modifying into "changes in DOC color", as it most accurately describes the approach used here.*

*P1 L17: "Photo-chemistry" includes all the chemical effects of light, so that is not incorrect, but, as a "dominant process in DOC transformation in the epilimnia", do the authors specifically mean "photo-decay" or "photo-degradation?*

*P1 L20: Would "moreover" be more appropriate than "instead"?*

*P2 L2: Consider changing "and to cause" into "and cause"*

*P3 L1: Consider changing "selected" into "selective"*

*P3 L28: absorbance or absorption coefficient?*

*P6 L27: Fig. A2 should be Fig. S2?*

*P7 L29: "was" should be "were"*

Reply: Changed as suggested

**Author's change: See changes on p. 1, l. 13; p. 1, l. 17; p. 1, l. 20; p. 2, l. 2; p. 3, l. 7; p. 4, l. 3; p. 7, l. 20; and p. 8, l. 30**

[revised manuscript text omitted]

Moved (insertion) [1]

Moved up [1]: Figure S3. Losses in dissolved organic carbon and in color during 2-week dark bioassays in 20 °C

**Text S1. Supplemental methods**

*Discharge measurements*

Daily discharge for the whole study periods 2012-2014 was extracted from hourly measurements performed for five of the study lakes, either at the outlet or at the inlet. Inlet measurements were re-scaled through multiplication by the ratio of 'outlet

5  catchment area' : 'inlet catchment area'. Water height loggers of model WT-HR 100 (Trutrack Inc., New Zealand) were placed at reaches with well-defined banks, and discharge was calculated using established rating curves based on the salt dilution method (number of observations per stream: 24-32, normalized root mean square error: 0.13-0.29). In the four lakes with the smallest catchments (Lapptjärn, Mångstrettjärn, Nästjärnen and Fisklösan), we considered inlet runoff to be too small to be assessed from water height. These lakes had no continuously flowing inlets and diffuse water flow may dominate during parts

10  of the year. For these lakes, we assumed that specific discharge was identical to that of the lake Övre Björntjärnen, and thus we rescaled the discharge from Övre Björntjärnen according to the catchment size of each of these four lakes. The assumption of similar specific runoff can be justified by the proximity of the lakes (see Table S1), the similar catchment slope gradient (ca 10%) and composition of bedrock (paleoproterozoic granitoids etc.), soils (peat and podzol soils on glacial till) and vegetation (coniferous forest and peatlands).

15  Prior to 2012, discharge for all sampled lakes was approximated from specific discharge from the stream Kallkällsbäcken in the Krycklan catchment (50 km northeast of Övre Björntjärnen), where stream water levels have been recorded continuously since 1980 using a pressure transducer and a 90° V-notch weir housed in a heated shed (Laudon et al. 2013). Discharge measured at the Övre Björntjärnen inlet, 1996-1998 (Jansson et al. 2001), indicated that the mean specific discharge (approximately 10 L s-1 km-2) was not significantly different from that recorded in the Krycklan catchment over

20  the same period (Köhler et al. 2008). Moreover, manually registered water levels at the inlet of Övre Björntjärnen on a total of 18 dates, 2007-2009, demonstrated a strong correlation with the water level in the Krycklan stream ($R^2 = 0.85$, n = 18, p < 0.01).

References to supplemental methods Text S1

25  Jansson, M., A. K. Bergström, S. Drakare, and P. Blomqvist. 2001. Nutrient limitation of bacterioplankton and phytoplankton in humic lakes in northern Sweden. Freshwat. Biol. 46: 653-666.

Köhler, S. J., I. Buffam, H. Laudon, and K. H. Bishop. 2008. Climate's control of intra-annual and interannual variability of total organic carbon concentration and flux in two contrasting boreal landscape elements. J. Geophys. Res. Biogeosciences 113: G03012.

30  Laudon, H., I. Taberman, A. Agren, M. Futter, M. Ottosson-Lofvenius, and K. Bishop. 2013. The Krycklan Catchment Study-A flagship infrastructure for hydrology, biogeochemistry, and climate research in the boreal landscape. Water Resour. Res. 49: 7154-7158.

**Table S1. Statistical details for linear regressions of dissolved organic carbon (DOC) properties as functions of water transit time (yrs).** The regressions are used in Fig. 2 and partly in Fig. 3 of the main paper. Full data plots are shown in Fig. S5. Columns from left to right depict the name of lake, water layer ('Epi' for epilimnion and 'hypo' for hypolimnion), response (y) variable, slope ± standard error, y-axis intercept, number of observations (n), explained variance ($R^2$) and 2-tailed significance of the slope (p).

| Site name | Layer | y | Slope ± SE | Intercept | n | $R^2$ | p |
|---|---|---|---|---|---|---|---|
| Fisklösan | Epi | $a_{254} : a_{365}$ | 0.42 ± 0.17 | 4.15 | 38 | 0.14 | 0.019 |
| Nästjärnen | Epi | $a_{254} : a_{365}$ | 0.45 ± 0.13 | 3.10 | 40 | 0.23 | 0.002 |
| Mångstrettjärn | Epi | $a_{254} : a_{365}$ | 0.11 ± 0.06 | 3.97 | 41 | 0.08 | 0.081 |
| Lapptjärn | Epi | $a_{254} : a_{365}$ | 0.20 ± 0.08 | 3.98 | 41 | 0.14 | 0.015 |
| Lillsjöliden | Epi | $a_{254} : a_{365}$ | -0.08 ± 0.19 | 4.11 | 35 | 0.00 | 0.696 |
| Nedre Björntjärnen | Epi | $a_{254} : a_{365}$ | -0.11 ± 0.09 | 4.05 | 59 | 0.03 | 0.220 |
| Struptjärnen | Epi | $a_{254} : a_{365}$ | -0.22 ± 0.10 | 4.10 | 38 | 0.14 | 0.027 |
| Övre Björntjärnen | Epi | $a_{254} : a_{365}$ | -0.41 ± 0.14 | 4.11 | 72 | 0.11 | 0.004 |
| Stortjärnen | Epi | $a_{254} : a_{365}$ | -0.50 ± 0.07 | 4.02 | 31 | 0.44 | 0.000 |
| Fisklösan | Epi | $a_{420} : DOC$ | -0.06 ± 0.04 | 0.38 | 38 | 0.07 | 0.113 |
| Nästjärnen | Epi | $a_{420} : DOC$ | -0.08 ± 0.03 | 0.60 | 40 | 0.19 | 0.005 |
| Mångstrettjärn | Epi | $a_{420} : DOC$ | -0.02 ± 0.02 | 0.51 | 41 | 0.02 | 0.368 |
| Lapptjärn | Epi | $a_{420} : DOC$ | -0.04 ± 0.04 | 0.51 | 41 | 0.03 | 0.315 |
| Lillsjöliden | Epi | $a_{420} : DOC$ | -0.03 ± 0.06 | 0.50 | 35 | 0.00 | 0.662 |
| Nedre Björntjärnen | Epi | $a_{420} : DOC$ | 0.08 ± 0.05 | 0.55 | 59 | 0.05 | 0.089 |
| Struptjärnen | Epi | $a_{420} : DOC$ | -0.02 ± 0.06 | 0.56 | 38 | 0.00 | 0.716 |
| Övre Björntjärnen | Epi | $a_{420} : DOC$ | 0.05 ± 0.08 | 0.54 | 72 | 0.01 | 0.542 |
| Stortjärnen | Epi | $a_{420} : DOC$ | 0.15 ± 0.05 | 0.57 | 31 | 0.26 | 0.004 |
| Fisklösan | Epi | $a_{420}$ (m$^{-1}$) | -0.64 ± 0.28 | 3.07 | 38 | 0.12 | 0.028 |
| Nästjärnen | Epi | $a_{420}$ (m$^{-1}$) | -0.69 ± 0.23 | 4.85 | 40 | 0.19 | 0.004 |
| Mångstrettjärn | Epi | $a_{420}$ (m$^{-1}$) | -0.80 ± 0.33 | 6.70 | 41 | 0.13 | 0.021 |
| Lapptjärn | Epi | $a_{420}$ (m$^{-1}$) | -1.80 ± 0.57 | 7.54 | 41 | 0.20 | 0.003 |
| Lillsjöliden | Epi | $a_{420}$ (m$^{-1}$) | 1.61 ± 1.88 | 7.30 | 35 | 0.01 | 0.395 |
| Nedre Björntjärnen | Epi | $a_{420}$ (m$^{-1}$) | 2.00 ± 1.22 | 10.08 | 59 | 0.05 | 0.106 |
| Struptjärnen | Epi | $a_{420}$ (m$^{-1}$) | 2.75 ± 2.47 | 10.81 | 38 | 0.04 | 0.273 |
| Övre Björntjärnen | Epi | $a_{420}$ (m$^{-1}$) | 3.04 ± 2.74 | 11.32 | 72 | 0.02 | 0.270 |
| Stortjärnen | Epi | $a_{420}$ (m$^{-1}$) | 4.91 ± 1.04 | 11.27 | 31 | 0.27 | 0.000 |
| Fisklösan | Epi | DOC (mg L$^{-1}$) | -0.48 ± 0.56 | 8.18 | 38 | 0.02 | 0.389 |
| Nästjärnen | Epi | DOC (mg L$^{-1}$) | -0.36 ± 0.27 | 8.76 | 40 | 0.04 | 0.192 |
| Mångstrettjärn | Epi | DOC (mg L$^{-1}$) | -1.22 ± 0.33 | 13.23 | 41 | 0.26 | 0.001 |
| Lapptjärn | Epi | DOC (mg L$^{-1}$) | -2.68 ± 0.56 | 14.86 | 41 | 0.37 | 0.000 |
| Lillsjöliden | Epi | DOC (mg L$^{-1}$) | 1.88 ± 5.13 | 15.31 | 35 | 0.00 | 0.717 |
| Nedre Björntjärnen | Epi | DOC (mg L$^{-1}$) | 1.87 ± 3.00 | 18.31 | 59 | 0.01 | 0.536 |
| Struptjärnen | Epi | DOC (mg L$^{-1}$) | 4.71 ± 4.63 | 19.85 | 38 | 0.03 | 0.317 |
| Övre Björntjärnen | Epi | DOC (mg L$^{-1}$) | 2.55 ± 6.55 | 21.39 | 72 | 0.00 | 0.698 |
| Stortjärnen | Epi | DOC (mg L$^{-1}$) | 4.77 ± 2.11 | 18.97 | 31 | 0.15 | 0.031 |

Formatted Table

| | | | | | | | |
|---|---|---|---|---|---|---|---|
| Fisklösan | Hypo | $a_{254} : a_{365}$ | 0.11 ± 0.21 | 4.13 | 28 | 0.01 | 0.624 |
| Nästjärnen | Hypo | $a_{254} : a_{365}$ | 0.04 ± 0.09 | 3.68 | 30 | 0.01 | 0.675 |
| Mångstrettjärn | Hypo | $a_{254} : a_{365}$ | -0.04 ± 0.10 | 3.84 | 31 | 0.00 | 0.732 |
| Lapptjärn | Hypo | $a_{254} : a_{365}$ | -0.01 ± 0.15 | 3.85 | 31 | 0.00 | 0.926 |
| Lillsjöliden | Hypo | $a_{254} : a_{365}$ | 0.05 ± 0.20 | 3.85 | 33 | 0.00 | 0.799 |
| Nedre Björntjärnen | Hypo | $a_{254} : a_{365}$ | -0.23 ± 0.13 | 4.06 | 29 | 0.10 | 0.087 |
| Struptjärnen | Hypo | $a_{254} : a_{365}$ | -0.01 ± 0.17 | 3.83 | 29 | 0.00 | 0.934 |
| Övre Björntjärnen | Hypo | $a_{254} : a_{365}$ | -0.26 ± 0.12 | 4.02 | 45 | 0.10 | 0.031 |
| Stortjärnen | Hypo | $a_{254} : a_{365}$ | -0.62 ± 0.10 | 4.07 | 29 | 0.58 | 0.000 |
| Fisklösan | Hypo | $a_{420} : DOC$ | 0.02 ± 0.06 | 0.39 | 28 | 0.00 | 0.776 |
| Nästjärnen | Hypo | $a_{420} : DOC$ | 0.03 ± 0.04 | 0.46 | 30 | 0.01 | 0.538 |
| Mångstrettjärn | Hypo | $a_{420} : DOC$ | 0.11 ± 0.05 | 0.40 | 31 | 0.14 | 0.037 |
| Lapptjärn | Hypo | $a_{420} : DOC$ | 0.09 ± 0.07 | 0.54 | 31 | 0.05 | 0.249 |
| Lillsjöliden | Hypo | $a_{420} : DOC$ | -0.06 ± 0.07 | 0.56 | 33 | 0.02 | 0.424 |
| Nedre Björntjärnen | Hypo | $a_{420} : DOC$ | 0.19 ± 0.10 | 0.49 | 29 | 0.11 | 0.072 |
| Struptjärnen | Hypo | $a_{420} : DOC$ | -0.02 ± 0.07 | 0.62 | 29 | 0.00 | 0.776 |
| Övre Björntjärnen | Hypo | $a_{420} : DOC$ | 0.17 ± 0.05 | 0.50 | 45 | 0.20 | 0.002 |
| Stortjärnen | Hypo | $a_{420} : DOC$ | 0.20 ± 0.05 | 0.54 | 29 | 0.42 | 0.000 |
| Fisklösan | Hypo | $a_{420}$ (m$^{-1}$) | -0.16 ± 0.53 | 3.26 | 28 | 0.00 | 0.770 |
| Nästjärnen | Hypo | $a_{420}$ (m$^{-1}$) | -0.42 ± 0.56 | 7.11 | 30 | 0.02 | 0.453 |
| Mångstrettjärn | Hypo | $a_{420}$ (m$^{-1}$) | 0.4 ± 0.97 | 7.48 | 31 | 0.01 | 0.679 |
| Lapptjärn | Hypo | $a_{420}$ (m$^{-1}$) | -0.21 ± 1.51 | 8.89 | 31 | 0.00 | 0.889 |
| Lillsjöliden | Hypo | $a_{420}$ (m$^{-1}$) | -1.04 ± 0.84 | 9.79 | 33 | 0.05 | 0.227 |
| Nedre Björntjärnen | Hypo | $a_{420}$ (m$^{-1}$) | 3.33 ± 1.37 | 9.67 | 29 | 0.18 | 0.022 |
| Struptjärnen | Hypo | $a_{420}$ (m$^{-1}$) | 3.12 ± 3.25 | 13.98 | 29 | 0.03 | 0.345 |
| Övre Björntjärnen | Hypo | $a_{420}$ (m$^{-1}$) | 3.83 ± 1.12 | 10.81 | 45 | 0.21 | 0.001 |
| Stortjärnen | Hypo | $a_{420}$ (m$^{-1}$) | 17.7 ± 2.57 | 5.40 | 29 | 0.64 | 0.000 |
| Fisklösan | Hypo | DOC (mg L$^{-1}$) | -0.16 ± 0.53 | 8.25 | 28 | 0.10 | 0.110 |
| Nästjärnen | Hypo | DOC (mg L$^{-1}$) | -0.42 ± 0.56 | 13.95 | 30 | 0.06 | 0.205 |
| Mångstrettjärn | Hypo | DOC (mg L$^{-1}$) | 0.4 ± 0.97 | 16.74 | 31 | 0.13 | 0.044 |
| Lapptjärn | Hypo | DOC (mg L$^{-1}$) | -0.21 ± 1.51 | 15.67 | 31 | 0.14 | 0.075 |
| Lillsjöliden | Hypo | DOC (mg L$^{-1}$) | -1.04 ± 0.84 | 17.28 | 33 | 0.00 | 0.831 |
| Nedre Björntjärnen | Hypo | DOC (mg L$^{-1}$) | 3.33 ± 1.37 | 19.40 | 29 | 0.00 | 0.747 |
| Struptjärnen | Hypo | DOC (mg L$^{-1}$) | 3.12 ± 3.25 | 22.55 | 29 | 0.08 | 0.132 |
| Övre Björntjärnen | Hypo | DOC (mg L$^{-1}$) | 3.83 ± 1.12 | 21.60 | 45 | 0.00 | 0.888 |
| Stortjärnen | Hypo | DOC (mg L$^{-1}$) | 17.7 ± 2.57 | 12.19 | 29 | 0.56 | 0.000 |

[Figure]

**Figure S1. Losses in dissolved organic carbon (DOC) and in color (absorbance at 420 nm) during 2-week dark bioassays in 20 °C** performed on ambient and nitrogen-amended water from Nedre Björntjärnen (four replicate dates representing different seasons). If N was limiting the DOC processing rates *in situ*, there should have been an impact by N addition also in these short-term laboratory experiments. Bars and error bars show mean + standard error of four dates prior to the whole-lake nutrient manipulation (see methods in main paper for more information).

[Figure]

**Figure S2. Means (solid lines) ± 1 *SD* (dotted lines) of hydrological variables describing (A) lake mixing, (B) catchment discharge and (C-D) water transit time (WTT) as functions of time of year** (month of year). Note that B-D show mean and *SD* derived from the log-normal distribution of y-axis values. The variables were assessed separately for each lake and study year, but for illustrative purposes the figure shows the average of 38 linearly interpolated curves obtained from 9 lakes in northern Sweden during 3-7 years of measurement per lake. The hypolimnetic WTT (in D) is only shown for periods during which the hypolimnion was typically present.

[Figure]

[Figure]

**Figure S3.** Comparisons between absorbance ratio $a_{254} : a_{365}$ and the measured concentrations of low molecular weight carbon compounds (LMWC). (a-b) Timeline plots for $a_{254} : a_{365}$ and absolute LMWC concentrations shown for two sites in the Björntjärnarna catchment (a and b, respectively) from April to October 2009. Significant correlations between the two variables are indicated by the statistics below each curve pair. (c-d) Linear relationships between $a_{254} : a_{365}$ and (c) relative or (d) absolute LMWC concentrations in the pooled dataset of all observations. The LMWC represents a sum of 39 of the most common organic acids, free amino acids and simple carbohydrates. Note that panels a-b share the same y axes.

[Figure]

[Figure]

**Figure S4. Discharge (line, right y axis) and dissolved organic carbon properties (symbols, left y axis) plotted over Gregorian calendar time for the Björntjärnarna catchment in northern Sweden.** The primary y-axis variables are (A) ratio between the absorbance at the wavelengths of 254 nm and 365 nm, (B) ratio between absorbance at 420 nm and DOC, (C) dissolved organic carbon and (D) absorbance at 420 nm (n = 260, study years 2006-2014).

[Figure]

**Figure S5 (previous page). Raw data plots for linear regressions\* detailed in Table S1 and used in Figs. 2-3 of the main paper.** (a) The upper half of figure shows epilimnetic data ('epi' sites) and (b) the lower half shows hypolimnetic data ('hyp' sites).

\*Significance: solid line, $p < 0.01$; dashed line, $p < 0.05$. See table S1 for details.